# EQUALS: An Audio-Visual LLM with One-Stage Question-Guided Alignment and Flexible Fusion

## Abstract

Audio-Visual Question Answering (AVQA) has emerged as a crucial task for multimodal reasoning in human-computer interaction, requiring models to align and interpret visual and auditory signals conditioned on natural language questions. Despite recent progress, three key challenges remain: (1) difficulty in locating question-relevant segments within lengthy and redundant video streams, (2) suboptimal audio-visual alignment due to the decoupling between pretraining and task-specific supervision, and (3) insufficient flexibility in fusion strategies across diverse tasks. To address these issues, we propose **EQUALS** (on**E**-stage **Q**uestion g**U**ided **A**lignment and f**L**exible fu**S**ion), a unified end-to-end AVQA framework. **EQUALS** integrates compression, alignment, and fusion within a single stage. Specifically, we interleave optimal transport-based loss modules before and after the question-guided pooling module to achieve fine-grained semantic alignment. To enhance adaptability in fusion, we introduce **FlexFuseMoE**, a mixture-of-experts module that supports early, mid, and late fusion via flexible expert routing. Experiments on MUSIC-AVQA and its challenging variant FortisAVQA demonstrate that EQUALS achieves new state-of-the-art results with interpretability. Our findings highlight the importance of jointly modeling alignment and fusion under explicit question guidance, offering a flexible and scalable solution for audio-visual understanding.

## 1 Introduction

Audio-Visual Question Answering (AVQA) Yun et al. (2021); Li et al. (2022); Yang et al. (2022); Liu et al. (2024b); Ma et al. (2024) has emerged as a central problem for natural human–computer interaction, attracting increasing attention in recent years Han et al. (2024); Lin et al. (2024); Zhou et al. (2025); Bravo et al. (2025). The task requires integrating auditory and visual signals in a question-aware manner, which is particularly challenging in realistic acoustic environments involving speech, music, and ambient sounds. Progress in this area has been largely driven by the development of benchmarks: compared with AVQA Yang et al. (2022) and Pano-AVQA Yun et al. (2021), which emphasize object-level semantics and panoramic reasoning, the MUSIC-AVQA series Li et al. (2022); Liu et al. (2024b); Ma et al. (2024; 2025) introduces fine-grained, instance-level annotations in dynamic musical performances, enabling richer question types such as comparative, temporal, and counting, and providing a comprehensive testbed for spatiotemporal reasoning.

Despite these advances, several challenges remain. First, many approaches adopt a two-stage paradigm that combines contrastive learning with masked reconstruction Gurram et al. (2022); Wang et al. (2024); Liu et al. (2024a); Araujo et al. (2025); Wang et al. (2025b); Cheng et al. (2025); while effective for representation learning, these methods decouple alignment from downstream supervision, require complex pretraining, and hinder joint optimization. In contrast, Optimal Transport (OT) Santambrogio (2015) has proven effective for multi-source alignment Zhu & Luo (2024); Yan et al. (2024); Rho et al. (2025); Wang & Zhao (2025); Wang et al. (2025a), with empirical studies confirming its superiority over contrastive paradigms Chowdhury et al. (2024). Second, long video sequences contain high redundancy, with only a small fraction relevant to the query; while recent methods Ye et al. (2024); Li et al. (2024); Diao et al. (2025); Kim et al. (2025) introduce question-aware mechanisms, they often compress raw streams without explicitly modeling

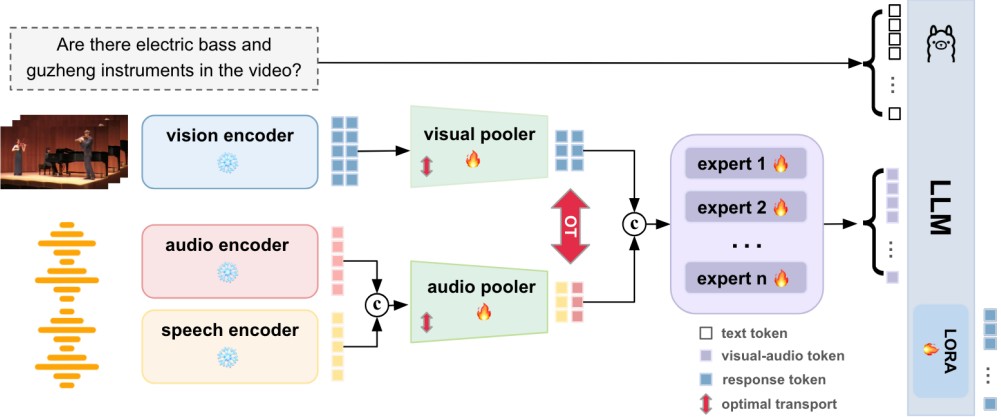

Figure 1: Overview of the proposed framework, which consists of three key components: (1) Question-guided Pooling for extracting question-aware modality features; (2) OTLoss to enforce cross-modal alignment via optimal transport before and after pooling operation; and (3) FlexFuse-MoE, a task-dependent mixture-of-experts for adaptive multimodal fusion.

question–modality alignment. Third, fusion design remains insufficiently adaptive: early Shi et al. (2024), mid-level Yang et al. (2024a); Awan et al. (2024); Zhou et al. (2025), and late Senocak et al. (2023); Li et al. (2024); Ryumina et al. (2024); Bravo et al. (2025) strategies lack flexibility to dynamically balance modalities, while recent Mixture-of-Experts (MoEs) Han et al. (2024); Lin et al. (2024); Kim et al. (2025) introduce conditional computation and capacity allocation but remain restricted to vanilla architectures.

These limitations highlight the need for AVQA models that jointly optimize fine-grained cross-modal alignment under task supervision, explicitly leverage question semantics to filter redundancy, and incorporate adaptive fusion mechanisms that allocate capacity where needed. Addressing these challenges is essential to advance robust and generalizable audio-visual reasoning.

To address these limitations, and motivated by how humans selectively attend to content with a question in mind a unified single-stage audio-visual LLM framework that integrates compression, alignment, and fusion in an end-to-end manner. Given an audio-visual question, EQUALS first employs a plug-in optimal transport loss module (OTLoss) to align the question with audio and visual streams independently. A Question-Guided Pooling module is then applied to extract semantically relevant features, directed by the question. After compression, OTLoss is applied again to further refine audio-visual alignment, forming an interleaved *align–compress–align* pipeline that yields a more consistent representation under question guidance. Finally, a task-adaptive **FlexFuse-MoE** module leverages dynamic routing and diverse expert combinations to effectively integrate multiple fusion strategies. The resulting features are fed into a large language model for reasoning. Our contributions are threefold: (1) We propose a unified training framework that interleaves optimal transport–based alignment before and after pooling. This design ensures semantically consistent compression by explicitly aligning question–audio/visual content and further enforcing bidirectional alignment of compressed audio-visual streams, leading to compact and coherent multimodal representations. (2) We introduce a linguistically guided pooling module that selectively compresses question-relevant features, enabling focused multimodal representations conditioned on the question. (3) We present **FlexFuseMoE**, a task-adaptive mixture-of-experts fusion module that dynamically integrates early, mid, and late fusion strategies via diverse expert routing, substantially enhancing reasoning capability over audio-visual inputs. We defer an extended discussion of related work to Appendix A.

## 2 METHOD

We propose a unified framework for question-guided multimodal representation learning that jointly addresses alignment, compression, and fusion within a single-stage architecture. Unlike prior two-stage approaches that decouple pretraining from downstream supervision, our design leverages the

question as a guiding signal to adaptively extract task-relevant information from both audio and video modalities, thereby enabling efficient and question-aware multimodal fusion. The framework is built upon three key components. First, *Question-Guided Pooling* generates modality representations that are explicitly conditioned on the question. Second, an interleaved *Optimal Transport Loss (OTLoss)* module enforces fine-grained cross-modal alignment, serving as a plug-in objective directly coupled with task supervision. Third, *FlexFuseMoEs*, an advanced flexible mixture-of-experts mechanism, perform dynamic and adaptive fusion of multimodal features by allocating capacity according to question-specific demands. An overview of the proposed architecture is illustrated in Figure 1.

## 2.1 TASK DEFINATION

Given a multimodal input $X = \{x^{\mathrm{a}}, x^{\mathrm{v}}\}$ and a natural language question $Q$, our goal is to learn a question-aware fused representation that can be used for downstream tasks such as audio-visual question answering. We dynamically condition the compressive pooling process on the semantic content of $Q$, reducing redundant information and mitigating interference during the fusion stage.

## 2.2 QUESTION-GUIDED POOLING

In this section, we introduce our Question-Guided Pooling design, which leverage task-specific descriptive questions as guidance to extract information from audio and video modalities using 2D and 3D attention mechanisms, respectively. Figure 2 describe the details of the module process.

### 2.2.1 QUESTION GLOBAL SEMANTIC

We begin by encoding the natural language question $Q$ using a pre-trained BERT model Radford et al. (2021a). The [CLS] token is adopted as the global semantic representation $h_Q^{\mathrm{bert[CLS]}} \in \mathbb{R}^d$, where $d$ denotes the embedding dimensionality. We use BERT rather than the embeddings from the first layer of LLM, since the latter capture only local token-level features and fail to provide a coherent global representation of the entire question.

### 2.2.2 QUESTION-GUIDED POOLING

Given audio inputs $x^a$ and video inputs $x^v$, we first extract modality-specific representations using pretrained encoders:

$$
\begin{aligned}
h^a &= \mathrm{Enc}_a(x^a) \in \mathbb{R}^{T_a \times B \times D_a}, \\
h^v &= \mathrm{Enc}_v(x^v) \in \mathbb{R}^{T_v \times H \times W \times D_v},
\end{aligned}
\tag{1}
$$

where $T_a(T_v), F_a(F_v), B(H, W)$ and $D$ represent the temporal, spatial, and feature dimensions of audio and video, respectively. We then flatten the frequency(spatial) and temporal dimensions into sequences, applying a $\phi(\cdot)$ operation (two-layer MLP with ReLU activation and output dimension of LLaMA3) to obtain the features prior modality-specific pooling:

$$
\begin{aligned}
\tilde{h}^a &= \phi(\mathrm{Flatten}(h_a)), & \tilde{h}^a &\in \mathbb{R}^{T_a' \times D_{\mathrm{llama}}}, \; T_a' = T_a \times B, \\
\tilde{h}^v &= \phi(\mathrm{Flatten}(h_v)), & \tilde{h}^v &\in \mathbb{R}^{T_v' \times D_{\mathrm{llama}}}, \; T_v' = T_v \times H \times W,
\end{aligned}
\tag{2}
$$

In order to effectively guide the pooling process with global question semantics, we project the question embedding $(h_{\mathrm{[CLS]}}^{\mathrm{BERT}})$ into the hidden spaces of both the audio and visual modalities via a learnable linear transformation , resulting in the projected embedding $h_{\mathrm{q}}$. The attention weights over the modality features are computed using a softmax-based attention mechanism, as follows:

$$h_q = \text{Linear}(h_{[\text{CLS}]}^{\text{BERT}}),$$

$$\alpha_t^a = \text{softmax}\left(\frac{h_q^\top W_a \, \tilde{h}_t^a}{\sqrt{d}}\right), \quad t = 1, \ldots, T_a',$$

$$\alpha_t^v = \text{softmax}\left(\frac{h_q^\top W_v \, \tilde{h}_t^v}{\sqrt{d}}\right), \quad t = 1, \ldots, T_v'. \tag{3}$$

where $\alpha_{a,t}$ and $\alpha_{v,t}$ represent the attention weight for the t-th feature, $h_{a,t}$ and $h_{v,t}$ are the reshaped features piror pooling operation, and $W_a$ and $W_v$ are the learned weight matrix for respective modality stream. This attention mechanism ensures that features most relevant to the question semantics are prioritized during the pooling operation. Finally, the compressed and informative representations $\hat{h}_{pooled}^a$ and $h_{pooled}^v$ are obtained by taking an weighted average pooling on features $\hat{h}_t^a$ and $\hat{h}_t^v$ that are enhanced by the text-to-audio/video alignment (see Sec. 2.3.1) on $\tilde{h}_t^a$ and $\tilde{h}_t^v$ respectively:

$$h_{\text{pooled}}^a = \text{AttnAvgPool}\left(\left\{\alpha_t^a \, \tilde{h}_t^a\right\}_{t=1}^{T_a'}\right),$$

$$h_{\text{pooled}}^v = \text{AttnAvgPool}\left(\left\{\alpha_t^v \, \tilde{h}_t^v\right\}_{t=1}^{T_v'}\right). \tag{4}$$

## 2.3 OPTIMAL TRANSPORT BASED ALIGNMENT CROSS MODALITIES

We design a family of Optimal Transport Loss (OTLoss) modules that enable flexible alignment across arbitrary modality pairs. Concretely, we develop three task-specific variants. *OTLoss-AT* (audio–text) and *OTLoss-VT* (video–text) transfer question semantics into the local "patch" embeddings of audio or video, thereby enhancing question-aware representation learning. *OTLoss-AV* (audio–video) focuses on aligning complementary audio–visual information after ququestion-guided compression, preparing more informative representations for the subsequent fusion stage. To support diverse learning dynamics, these modules are interleaved at different stages of fine-tuning. Furthermore, to prevent spurious alignments caused by variable-length inputs, we modify the transport cost by imposing a large penalty on distances involving padding tokens, thereby ensuring that the transport plan is concentrated on semantically meaningful regions.

### 2.3.1 TEXT-TO-AUDIO/VIDEO ALIGNMENT BEFORE QUESTION-GUIDED POOLING

As shown in Figure 2, to align textual information with acoustic and visual streams, we adopt an Optimal Transport (OT) formulation that treats question tokens as queries and audio/video tokens as candidates for alignment. Specifically, We use LLaMA3Grattafiori et al. (2024) to encode the input question as $h^l \in \mathbb{R}^{T_l \times D_{llama}}$ due to its fine-grained token-level representations and strong semantic modeling capability. To match query embedding with the dimensionality of audio and video encoders, a linear layer is applied to project the text embedding into audio and video's hidden dimensions respectively.

As mentioned in Section 3.2.2, Given the flattened audio representation $\tilde{h}^a \in \mathbb{R}^{T_a' \times D_{llama}}$ and $h_l \in \mathbb{R}^{T_l \times D_{llama}}$, we first construct the cost matrix via cosine distance:

$$C_{ij}^{l2a} = 1 - \cos(h_l^i, h_a^j), \quad C_{ij}^{l2v} = 1 - \cos(h_l^i, h_v^j), \tag{5}$$

where $i$ indexes text tokens and $j$ indexes audio or visual tokens.

To obtain a soft assignment, we normalize each row to form a distribution over the target modality:

$$M_{i\cdot}^{l2a} = \tfrac{1}{T_l} \text{softmax}\left(-\tfrac{1}{\tau_a} C_{i\cdot}^{l2a}\right), \quad M_{i\cdot}^{l2v} = \tfrac{1}{T_l} \text{softmax}\left(-\tfrac{1}{\tau_v} C_{i\cdot}^{l2v}\right), \tag{6}$$

where $\tau_a, \tau_v$ are temperature parameters controlling the sharpness of the alignment (smaller $\tau$ leads to harder assignments). Alternatively, sparse operators such as sparsemax or entmax can be used, and a straight-through estimator (STE) can approximate hard $\arg\min$ assignments.

The resulting one-sided OT losses are formulated as:

$$\mathcal{L}_{\text{OT}}^{l \to a} = \langle M^{l2a}, C^{l2a} \rangle + \varepsilon_a H(M^{l2a}) + \alpha_a \left\| M^{l2a}\mathbf{1} - \frac{1}{T_l}\mathbf{1} \right\|_2^2, \tag{7}$$

$$\mathcal{L}_{\text{OT}}^{l \to v} = \langle M^{l2v}, C^{l2v} \rangle + \varepsilon_v H(M^{l2v}) + \alpha_v \left\| M^{l2v}\mathbf{1} - \frac{1}{T_l}\mathbf{1} \right\|_2^2, \tag{8}$$

where $\langle A, B \rangle = \sum_{ij} A_{ij} B_{ij}$, $H(M) = -\sum_{ij} M_{ij} \log M_{ij}$ is the entropy regularizer, and the last term softly enforces the row-sum constraint.

This formulation allows each text token to distribute its mass to semantically similar audio or visual tokens, producing a differentiable alignment map without extra parameters. The resulting one-sided OT alignment offers a lightweight yet effective way to guide cross-modal interaction, yielding question-augmented audio and video representations:

$$\hat{h}_a = \tilde{h}^a + M^{l2a}h_l, \quad \hat{h}_v = \tilde{h}^v + M^{l2v}h_l. \tag{9}$$

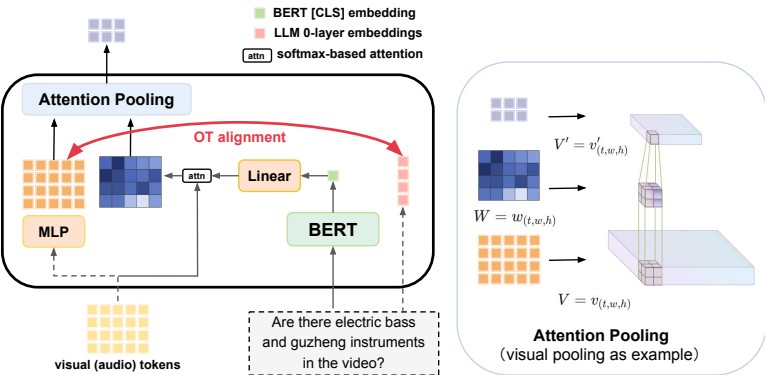

Figure 2: Question-Guided Pooling and Text-Audio/Video Alignment: Optimal transport-based alignment is first applied from text to audio or video modality using LLaMA-encoded question embeddings projected via an MLP block. Subsequently, for each modality, attention weights are computed through a softmax-based attention mechanism between the BERT [CLS] token and audio/video features to guide pooling. This ensures that semantically relevant regions are extracted based on the input question. Solid gray arrows indicate question-guided attention; dotted gray arrows represent optimal transport alignment; black arrows denote final attention pooling using the weight matrix (blue blocks) and aligned representation (orange blocks).

## 2.4 ALIGNMENT BETWEEN AUDIO AND VIDEO MODALITIES AFTER QUESTION-GUIDED POOLING

For audio–visual alignment, we focus on compressed patch-level features obtained after question-guided pooling, with the goal of aligning semantically corresponding information across modalities to prepare for the subsequent fusion stage. Given audio and video modalities $\mathcal{M}_{audio}$ and $\mathcal{M}_{video}$, we obtain $h_{\text{pooled}}^a$ and $h_{\text{pooled}}^v$ as their respective patch-level representations. Leveraging the semantic structure encoded in these features, we construct two discrete distributions $\theta_a \in P(\mathcal{H}_a)$ and $\theta_v \in P(\mathcal{H}_v)$ for the audio and video streams as follows:

$$\theta_a = \sum_{k=1}^{M} u_a(k)\,\delta_{f_a(k)}, \quad \theta_v = \sum_{l=1}^{N} u_v(l)\,\delta_{f_v(l)} \tag{10}$$

where $u_a \in \Delta^M$ and $u_v \in \Delta^N$ denote the weights of the corresponding probability distributions, which we set as uniform for simplicity, i.e., $u_a(k) = \frac{1}{M}$ and $u_v(l) = \frac{1}{N}$. The term $\delta_f$ denotes the

Dirac delta function placed at the supporting point $f$ in the embedding space where the distribution assigns non-zero probability Chen et al. (2022a).

The objective is to compute the optimal transport plan $\Omega$ that minimizes the transport cost between the two distributions $\theta_a$ and $\theta_v$, while preserving the underlying topological structure during cross-modal alignment. This is achieved by minimizing the Wasserstein Distance (WD), formally defined as:

$$\mathcal{L}_{\text{OT}}^{a \leftrightarrow v} = \mathcal{L}_{\text{OT}}(\theta_a, \theta_v) = \min_{\Omega \in \Psi(u_a, u_v)} \sum_{k=1}^{M} \sum_{l=1}^{N} \Omega_{kl} \, \phi(h_{pooled}^a(k), h_{pooled}^v(l)) \tag{11}$$

The set of feasible transport plans $(u_a, u_v)$ is defined as follows:

$$\Psi(u_a, u_v) = \left\{ \Omega \in \mathbb{R}_+^{M \times N} \,\middle|\, \Omega \mathbf{1}_N = u_a, \; \Omega^\top \mathbf{1}_M = u_v \right\} \tag{12}$$

and the ground cost function $\phi(f_a(k), f_v(l))$ is instantiated as the cosine distance between local patch-level embeddings from two modalities. An exact solution to the above formulation yields a sparse optimal transport plan $\Omega$, containing at most $2 \cdot \max(M, N) - 1$ non-zero entries. This sparsity property enhances both the interpretability and robustness of the resulting cross-modal alignment. Algorithmic detail on calculating WD is appended in Appendix B.

## 2.5 FLEXFUSEMOE

We propose **FlexFuseMoE**, a multimodal fusion module that dynamically routes features through a mixture-of-experts architecture. Building on DeepSeekMoE Dai et al. (2024), our model employs $N$ groups of $m$ fine-grained experts per layer ($K_s$ shared, $mN - K_s$ routed), which is different from vanilla MoE adopted for multimodal fusion in Han et al. (2024). Each expert is implemented as a feed-forward network:

$$E_k(x) = W_{2,k} \cdot \text{ReLU}(W_{1,k} x), \quad \text{for } k = 1, \dots, N - K_s \tag{13}$$

For each input token representation $f_t^l$ at layer $l$, the output combines contributions from always-active shared experts and selectively-activated routed experts:

$$h_t^l = \underbrace{\sum_{i=1}^{K_s} \text{FFN}_i(f_t^l)}_{\text{shared}} + \underbrace{\sum_{i=K_s+1}^{mN} g_{i,t} \, \text{FFN}_i(f_t^l)}_{\text{routed}} + f_t^l$$

$$g_{i,t} = \begin{cases} s_{i,t}, & \text{if } s_{i,t} \in \text{TopK}\big(\{s_{j,t}\}_{j=K_s+1}^{mN}, K_r = mK - K_s\big) \\ 0, & \text{otherwise} \end{cases} \tag{14}$$

$$s_{i,t} = \text{softmax}_i\big((f_t^l)^\top e_i^l\big)$$

$$e_i^l \sim \mathcal{N}(0, \sigma^2 I), \qquad \sigma = 0.001$$

The gating mechanism activates only the top-$K_r$ routed experts (default $K_r = 4$) based on their affinity scores $s_{i,t}$, while $K_s = 2$ shared experts process all tokens. This sparse activation maintains model capacity while ensuring computational efficiency. Learnable routing centroids $e_i^l$ enable automatic specialization of experts to different input patterns.

### ROUTING STRATEGIES

Inspired by FuseMoE Han et al. (2024), we implement three routing strategies:

- **Disjoint Routing(Disjoint Gates and Experts)** Each modality $m \in \{\text{audio}, \text{video}\}$ has independent gating networks $g_m$ and expert pools $E_m$, maximizing routing flexibility.

- **Joint Routing (Shared Gates and Experts)** A shared gating network $g$ processes all modalities through common experts $E$.
- **Per-Modality Routing (Shared Gates, Disjoint Experts)** A shared gating $g$ with modality-specific experts pool $E_m$.

Given modality-specific representations $f^a, f^v \in \mathbb{R}^{D_{llama}}$ obtained from question-guided pooling and alignment respectively, the fusion operation produces an output that preserves the original feature dimensionality, making it directly compatible with the downstream LLM architecture.

## 2.6 Total Loss

Our final training objective combines the cross-entropy loss for supervised tasks with the OT-based alignment losses:

$$\mathcal{L}_{\text{total}} = \mathcal{L}_{\text{CE}} + \lambda_{\text{at}}\mathcal{L}_{\text{OT-1s}}^{t \to v}(h_l, \tilde{h}^a) + \lambda_{\text{vt}}\mathcal{L}_{\text{OT-1s}}^{t \to v}(h_l, \tilde{h}^v) + \lambda_{\text{av}}\mathcal{L}_{\text{OT-1s}}^{a \leftrightarrow v}(h_{\text{pooled}}^a, h_{\text{pooled}}^v) \quad (15)$$

where $\lambda_{\text{at}}, \lambda_{\text{vt}}, \lambda_{\text{av}}$ are hyperparameters controlling the contribution of each OT loss component.

## 3 Experiments

We evaluate EQUALS on the widely used audio-visual benchmark MUSIC-AVQA and its more robust extension FortisAVQA. This dataset suite illustrates how audio-visual reasoning can be systematically studied under carefully designed conditions, while FortisAVQA further improves the testbed toward reducing biases and enabling more reliable evaluation. Additional details on datasets evolving and our preprocessing are provided in Appendix C.1 and C.2 respectively.

### 3.1 Datasets

**MUSIC-AVQA** Li et al. (2022) is a large-scale dataset for audio-visual question answering, containing 9k+ real and synthetic music performance videos across four instrument categories, with 45k QA pairs spanning audio, visual, and audio-visual modalities. Questions cover nine reasoning types (e.g., existential, counting, location), emphasizing spatiotemporal grounding of sounds and visual objects.

**FortisAVQA** Ma et al. (2024; 2025) extends MUSIC-AVQA with 200k+ paraphrased questions, increasing vocabulary by $5\times$ and improving linguistic diversity (e.g., variants of "Is the violin audible?"). It further introduces head/tail partitions by answer frequency with corresponding metrics, offering a more challenging and realistic benchmark for multimodal reasoning.

### 3.2 Experimental Settings

We implement our framework in PyTorch with DeepSpeed ZeRO-1 on $4\times$A100 GPUs under mixed-precision training. The question is encoded by **CLIP-ViT-L/14** Radford et al. (2021b), where the [CLS] token provides a global representation for pooling, while token-level features for text–audio/video alignment are obtained from **Llama3**. Videos are processed by **Intern-Video2** Wang et al. (2024) at 2 Hz into 256 spatial patches, and audio is represented via a dual-branch **BEATs** Chen et al. (2022b) and **Whisper v3** Radford et al. (2023) pipeline at 16 kHz. The backbone adopts **DeepSeek-MoE** Dai et al. (2024) with 16 routed experts and top-2 routing. Training uses Adam with cosine decay (init LR $3 \times 10^{-5}$), batch size 2, and uniform OT-based alignment loss weighting ($\lambda = 0.3$) across modality pairs. Full architectural and optimization details are provided in Appendix D.

### 3.3 Comparison on MUSIC-AVQA

We evaluate our method on the MUSIC-AVQA test split to validate its effectiveness. For comparison, we select nine representative baselines spanning small- and large-scale state-of-the-art methods Ma et al. (2025); Li et al. (2024). These include AudioQA models (FCNLSTM, CONVLSTM),

Table 1: Performance comparison on AVQA tasks of the MUSIC-AVQA dataset.

| Method | Audio | | | Visual | | | Audio-Visual | | | | | | Avg |
|---|---|---|---|---|---|---|---|---|---|---|---|---|---|
| | Count | Comp | Avg | Count | Local | Avg | Exist | Count | Local | Comp | Temp | Avg | |
| AVSD | 72.47 | 62.46 | 68.78 | 66.00 | 74.53 | 70.31 | 80.77 | 64.03 | 57.93 | 62.85 | 61.07 | 65.44 | 67.32 |
| PSTP-Net | 73.97 | 65.79 | 70.91 | 77.15 | 77.36 | 77.26 | 76.18 | 72.23 | 71.80 | 71.79 | 69.00 | 72.57 | 73.52 |
| LAViT | 74.27 | 65.06 | 70.86 | 69.89 | 77.12 | 73.55 | 81.21 | 62.03 | 65.93 | 60.90 | 63.96 | 66.78 | 69.29 |
| LAVisH | 78.18 | 58.74 | 70.98 | 75.65 | 78.75 | 77.21 | 81.41 | 63.54 | 71.98 | 57.76 | 66.38 | 68.30 | 71.13 |
| MAVEN | 79.44 | 54.10 | 72.79 | 80.49 | 93.50 | 86.99 | 87.00 | 66.67 | 73.85 | 54.95 | 68.24 | 69.94 | 74.60 |
| VITA | 59.81 | 45.90 | 54.76 | 50.41 | 34.96 | 42.68 | 54.00 | 49.46 | 46.92 | 27.93 | 41.18 | 43.74 | 45.44 |
| VideoLLaMA 2 | 79.44 | 52.46 | 69.64 | 81.30 | 82.93 | 82.11 | 77.00 | 63.44 | 77.69 | 59.46 | 64.71 | 68.98 | 72.56 |
| QWen2.5-VL | 48.60 | 55.00 | 51.80 | 55.28 | 53.66 | 54.47 | 44.00 | 52.17 | 63.57 | 37.84 | 41.18 | 47.75 | 50.14 |
| GPT-4o | 65.42 | 36.07 | 50.75 | 72.36 | 62.30 | 67.33 | 56.12 | 54.84 | 59.23 | 37.84 | 42.35 | 50.08 | 54.06 |
| **Ours(disjoint)** | 86.51 | 80.28 | 84.21 | 83.60 | **91.16** | 88.40 | 90.81 | 82.71 | 86.15 | 80.05 | 86.10 | 85.00 | 85.93 |
| **Ours(joint)** | **88.46** | **82.95** | **86.43** | 83.39 | 91.09 | 88.28 | 91.11 | 82.00 | **86.47** | **82.95** | **86.37** | **85.25** | **86.36** |
| **Ours(per-modality)** | 87.63 | 80.87 | 85.13 | **84.34** | 90.87 | **88.49** | **91.47** | **82.32** | 85.99 | 80.94 | 85.92 | 85.14 | 86.17 |

VideoQA models (HCRN, PSAC, HME), and AVQA models such as AVSD Alamri et al. (2019), LAViT Yun et al. (2021), and LAVisH Lin et al. (2023). We further compare against recent large multimodal models: MAVEN Ma et al. (2025), VideoLLaMA2 Cheng et al. (2024), Qwen2.5-VL Bai et al. (2023), and GPT-4o Hurst et al. (2024). Following common practice, Qwen2.5-VL and GPT-4o are evaluated in a zero-shot setting due to their closed-source nature and lack of explicit audio encoders, while MAVEN, VITA, and VideoLLaMA2 are fine-tuned on the benchmark for fair comparison. See additional details of baseline configurations in Appendix F.

As shown in Table 1, our model with joint fusion achieves a new state-of-the-art accuracy of 86.36, surpassing MAVEN by 11.76%. The gain is consistent across modalities: +13.64% on audio, +1.50% on visual, and +12.68% on audio-visual tasks. The improvement is most pronounced in audio (e.g., +14.19% over LAViT on comparison) and audio-visual reasoning (e.g., +11.16% on comparison and +17.37% on temporal tasks vs. PSTP-Net). By contrast, visual subtypes exhibit smaller margins, with +3.04% on counting (vs. VideoLLaMA2) and a slight decrease (-2.34%) on local (vs. MAVEN). These results highlight two key findings: (i) our framework effectively strengthens audio–visual integration, yielding robust improvements in modalities requiring cross-modal reasoning; and (ii) performance variation across question types confirms the adaptability of FlexFuseMoE in selecting appropriate fusion strategies under different task conditions.

## 3.4 ROBUSTNESS EVALUATION

As shown in Table 2, our method achieves state-of-the-art results across both head and tail splits. The disjoint fusion design improves prior bests by 17.2%, 5.98%, and 4.31% on location (head), comparison (head), and comparison (tail), respectively. The joint fusion variant further advances performance with +11.11%/+9.25% on location (tail/head) within visual and audio-visual modalities, and +3.89% on temporal (tail) under audio-visual. Per-modality fusion also yields large gains, including +11.91% and +32.88% on count/comparison in audio, +12.29% on count (tail) in visual, and +25.4% on temporal (head) in audio-visual. These consistent improvements demonstrate the effectiveness and cross-benchmark robustness of our design for diverse multimodal reasoning tasks.

Table 2: Performance comparison on FortisAVQA benchmark. (H: Head; T: Tail).

| Method | Audio QA | | | | Visual QA | | | | AVQA | | | | | | | | | | Avg |
|---|---|---|---|---|---|---|---|---|---|---|---|---|---|---|---|---|---|---|---|
| | Count | | Comp | | Count | | Local | | Exist | | Count | | Local | | Comp | | Temp | | |
| | H | T | H | T | H | T | H | T | H | T | H | T | H | T | H | T | H | T | |
| LAViT | 50.53 | 44.19 | 54.66 | 58.33 | 50.49 | 25.67 | 65.07 | 59.02 | 54.14 | 22.22 | 47.33 | 40.30 | 46.26 | 21.95 | 37.61 | 48.25 | 39.83 | 48.33 | 47.47 |
| LAVisH | 71.90 | 61.27 | 56.47 | 58.79 | 71.84 | 27.87 | 43.68 | 63.13 | 70.99 | 41.53 | 35.62 | 57.37 | 66.75 | 21.91 | 43.92 | 69.38 | 28.76 | 44.91 | 54.88 |
| MAVEN | 88.64 | 62.50 | 85.71 | 33.33 | 92.31 | 66.67 | 87.23 | 75.76 | 85.71 | 89.29 | 65.62 | 55.56 | 71.43 | 44.44 | 45.83 | 61.90 | 54.55 | 85.71 | 72.92 |
| VideoLLaMA2 | 89.47 | 67.44 | 64.00 | 40.00 | 84.95 | 67.57 | 55.02 | 74.59 | 87.97 | 44.44 | 61.33 | 52.24 | 70.48 | 51.22 | 59.83 | 62.28 | 49.15 | 68.33 | 66.85 |
| VITA | 91.45 | 48.28 | 62.67 | 35.00 | 74.03 | 38.89 | 15.74 | 46.19 | 63.16 | 45.30 | 34.34 | 37.29 | 47.22 | 27.27 | 77.78 | 28.95 | 27.27 | 49.50 | 48.66 |
| GPT-4o | 63.78 | 46.51 | 69.33 | 16.67 | 75.00 | 68.92 | 56.52 | 75.21 | 44.36 | 87.18 | 42.95 | 50.75 | 56.83 | 39.02 | 13.68 | 63.16 | 30.51 | 71.67 | 56.06 |
| QWen2.5-VL | 88.11 | 57.14 | 83.78 | 13.33 | 77.94 | 72.97 | 46.38 | 68.91 | 57.14 | 76.07 | 36.24 | 53.73 | 56.76 | 60.98 | 11.97 | 76.99 | 25.42 | 63.33 | 57.07 |
| Ours(disjoint) | 84.28 | **79.35** | 68.00 | 89.16 | 86.30 | 67.76 | **88.15** | 82.34 | **90.60** | 82.48 | **82.82** | 51.20 | 76.99 | 83.70 | **83.76** | 81.30 | 79.24 | 86.14 | 81.47 |
| Ours(joint) | 80.16 | **79.35** | 77.33 | 87.50 | 86.06 | 64.47 | 86.41 | **86.87** | 90.23 | **88.46** | 78.86 | 50.00 | **80.68** | **86.30** | 79.92 | 78.28 | 77.86 | **89.60** | **82.15** |
| Ours(per-mod.) | 81.96 | **79.35** | 78.67 | **91.67** | 85.58 | **85.26** | 87.05 | 82.34 | 87.97 | 60.68 | 74.01 | 50.00 | 78.53 | 82.22 | 76.50 | 60.00 | **79.59** | 84.16 | 79.30 |

While models such as LAViT and LAVisH perform competitively on MUSIC-AVQA, they degrade sharply on FortisAVQA, especially in the tail set (e.g., LAViT reaches only 22.22% on EXIST-type).

This suggests reliance on dataset-specific memorization rather than genuine multimodal reasoning. In contrast, our method and VideoLLaMA-2 exhibit much stronger robustness, with only 4.21% and 5.71% drops, respectively. Notably, large models in the zero-shot setting even match or surpass supervised results on MUSIC-AVQA, underscoring the strong generalization ability of large-scale multimodal LLMs.

## 3.5 ANALYSIS STUDIES

### 3.5.1 ATTENTIVE POOLING VS. AVERAGE POOLING

To evaluate the effectiveness of our Question-Guided Pooling (QGP), we compare it with average pooling (AP). QGP consistently outperforms AP on both MUSIC-AVQA and FortisAVQA, with overall accuracy gains of +1.03% and +1.92%, respectively. The improvements are particularly pronounced in audio-visual reasoning, e.g., +3.73% on *Existence* and +2.43% on *Counting* in MUSIC-AVQA, and up to +10.02% on compositional AVQA in FortisAVQA. These results highlight that leveraging question semantics for pooling preserves task-relevant cross-modal cues, whereas uniform averaging dilutes them. Complete results are provided in Appendix G.1. We further visualize the attention scores between question semantics and video frames over time, which provides additional evidence supporting the validity of our design, see in Appendix G.2.

### 3.5.2 OTLOSS ABLATION ANALYSIS

We conducted ablation study with our best average-performance models to verify the effectiveness of OTLoss module. The full OTLoss ($\lambda_{at} + \lambda_{vt} + \lambda_{av}$) achieves the best accuracy, with 86.17% on AVQA and 82.15% on FortisAVQA. Removing $\lambda_{av}$ leads to a drop of 2.37%/1.37%, while excluding $\lambda_{vt}$ and $\lambda_{at}$ results in a 1.83%/1.10% decrease, respectively. The baseline without OT losses performs worst (83.18%/79.90%). Similar trends are observed on FortisAVQA-Head and Tail subsets, where removing OT losses causes performance drops of 3.86% and 4.11%, respectively. These findings emphasize that the integration of cross-modal alignment through combined optimal losses leads to optimal performance, with the audio-text ($\lambda_{at}$) and video-text ($\lambda_{vt}$) losses proving to be particularly effective. Detailed breakdowns across subsets are reported in Appendix H.

### 3.5.3 KERNEL AND STRIDE CONFIGURATIONS

We additionally examine the impact of kernel and stride choices under the *joint routing* fusion strategy. This analysis aims to determine a balanced configuration that provides sufficient granularity while remaining computationally efficient. Comprehensive comparisons across different settings are reported in Appendix I.

### 3.5.4 IMPORTANCE OF AUDIO MODALITY IN INFERENCE

We further compare EQUALS with Qwen2.5-VL, a strong vision–language baseline. While Qwen2.5-VL performs well on purely visual queries, the absence of an audio modality leads to reduced confidence and errors on audio-dependent questions. EQUALS, by explicitly injecting audio and performing cross-modal alignment, consistently answers such queries correctly, showing that the gains arise from our architectural design rather than the language backbone alone. This underscores the importance of audio representations for robust audio–visual reasoning. Illustrative examples are provided in Appendix J.

## 4 CONCLUSION

We introduced EQUALS, a unified one-stage framework for audio-visual question answering that integrates question-guided alignment, feature compression, and flexible fusion. Experiments on MUSIC-AVQA and FortisAVQA show that EQUALS achieves state-of-the-art results across overall accuracy and fine-grained question types, validating its effectiveness and robustness. While current evaluation focuses on music-performance settings, future work will extend to more diverse audio-visual data to assess the generalizability of the framework.

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

## A   RELATED WORKS

### A.1   AUDIO-VISUAL UNDERSTANDING

Recent advancements in audio-visual learning emphasize spatio-temporal alignment and semantic fusion using advanced architectures. CAD Nadeem et al. (2024) employs contrastive learning to align audio-visual data across spatial, temporal, and semantic dimensions. SHMamba Yang et al. (2024b) utilizes hyperbolic geometry and an adaptive curvature module for dynamic cross-modal information exchange. The VALOR model Liu et al. (2024a) strengthens multi-modal associations through multi-modal grouping alignment and captioning tasks. Additionally, InternVideo Wang et al. (2022) and its enhanced versions Wang et al. (2024; 2025b) optimize spatiotemporal consis-tency via progressive training, improving performance on long-duration video tasks.

Question guidance has also been integral to the design of methods for audio-visual understanding. PSTP-Net Li et al. (2023) identifies relevant audio-visual regions to optimize feature selection and enhance comprehension of dynamic video content. MCD Ye et al. (2024) employs knowledge dis-tillation to align audio-visual and textual modalities, reducing the cross-modal semantic gap under question guidance. Recently, TWM model Diao et al. (2025) uses query-guided attention to focus on the most informative segments, retaining task-relevant information across temporal dimensions. QA-TIGER Kim et al. (2025) improves temporal dynamics by focusing on question-relevant frames with Gaussian modeling.

### A.2   MIXTURE OF EXPERTS IN MULTIMODAL LEARNING

Mixture-of-Experts (MoE) architectures have been increasingly adopted in multimodal learning for their ability to handle modality-specific representations while maintaining scalability. By activat-ing only a subset of expert modules conditioned on the input, MoE-based methods offer a flexible alternative to static fusion strategies, which often struggle with modality imbalance and redundancy.

Several recent studies have explored MoE in the context of multimodal fusion. MoPEJiang et al. (2024) introduces a prompt-based MoE design, where a set of learnable prompt experts are dynam-ically selected based on the input, enabling efficient adaptation with minimal parameter overhead. FuseMoEHan et al. (2024) addresses missing and irregularly sampled modalities using a Laplace gating mechanism, leading to improved robustness in time-series multimodal forecasting. QA-TIGER  Kim et al. (2025) applies a Mixture of Experts (MoE) to activate Gaussian models tailored to specific questions, enhancing temporal audio-visual reasoning by focusing on both consecutive and non-consecutive frames.

MoME Shen et al. (2024) incorprates sparsely gated vision and language experts to reduce task interference in general-purpose models.TTMESun et al. (2024), apply expert-based routing to tri-modal data during pretraining, showing improved performance in social media understanding tasks. In addition, While MIMoE-FNDLiu et al. (2025) adopts a hierarchical MoE structure with an interaction gating module to better model fine-grained correlations between text and visual content in multimodal fake news detection. These efforts together reflect the growing use of MoE in scenarios where modality interactions are complex and data distributions are heterogeneous.

## B  OTLoss Algorithms Math Details

In Algorithm1, we detail the computation of the Wasserstein distance used in our OTLoss. Given two modalities, the procedure initializes a scaled unity matrix and transport plan, constructs a cosine-based cost matrix, and applies an exponential decay to obtain a scaled similarity matrix. The Sinkhorn iteration is then performed by alternately updating the row and column scaling factors to refine the transport plan. Finally, the Wasserstein distance is computed via a Frobenius dot product between the cost matrix and the learned transport plan. This iterative process yields a stable and differentiable alignment map that can be seamlessly integrated into our framework for cross-modal optimization.

---

**Algorithm 1** EQUALS: Wasserstein Distance Computation in OTLoss

---

Modality 1: $\mathcal{M}1_{i\,i=1}^{k}$; Modality 2: $\mathcal{M}2_{j\,j=1}^{k}$ Total Optimal Transport steps: $\Omega$; Initial scaled unity matrix: $\sigma = \frac{1}{k}\mathbf{11}^{\top}$; Initial Transport Plan: $\Omega^{(1)} = \mathbf{11}^{\top}$; Cosine similarity matrix: $C_{ij} = c(I_i, A_j)$; similarity matrix decay factor: $\beta$; Scaled similarity matrix: $\tau_{ij} = e^{-\frac{C_{ij}}{\beta}}$. Learned Optimal Transport Plan: $\Omega$; Wasserstein Distance: $D_{\text{Wasserstein}}$.

**for** $t \in \{1, 2, 3, \cdots, \Omega\}$ **do** $\mathbf{Q} \leftarrow \tau \odot \Omega^{(t)}$ *$\odot$ is Hadamard product

 **for** $l \in \{1, 2, 3, \cdots, L\}$ **do** $\delta \leftarrow \frac{1}{k\mathbf{Q1}}$, $\sigma \leftarrow \frac{1}{k\mathbf{Q}^{\top}\delta}$  $\Omega^{(t+1)} \leftarrow \text{diag}(\delta)\mathbf{Q}\text{diag}(\sigma)$   $D_{\text{Wasserstein}} \leftarrow \langle \mathbf{C}^{\top}, \Omega \rangle$ *$\langle \cdot, \cdot \rangle$ is the Frobenius dot-product **return** $\Omega, D_{\text{Wasserstein}}$

---

## C  AVQA Datasets

### C.1  Datasets Evolution Description

To systematically evaluate the multimodal reasoning capacity of AVQA systems, a series of progressively refined datasets (You et al., 2025) in the context of musical instrument performance have been developed. The MUSIC-AVQA dataset (Li et al., 2022) serves as an early benchmark, introducing a large-scale collection of 9,288 videos (150+ hours) featuring performances across 22 instruments, with 45,867 question-answer (QA) pairs. These pairs are split into training (32,087 QA pairs), validation (4,595), and testing (9,185) sets, covering 33 question templates spanning 9 types (e.g., existential, location, counting, comparative, temporal). This dataset pioneered the evaluation of basic spatio-temporal and audio-visual understanding but revealed biases in certain question categories.

To address these limitations, MUSIC-AVQA v2.0 (Liu et al., 2024) rebalanced the dataset by adding 1,230 new videos (totaling 10,518) and generating 54,000 QA pairs. The splits were adjusted to 36,700 (training), 5,250 (validation), and 10,819 (testing) pairs, while ensuring no dominant answers exceeded 60% for binary questions or 50% for multi-class questions. This mitigation targeted biases in 15 templates, particularly improving representation in existential, counting, and comparative questions.

Further advancing robustness evaluation, MUSIC-AVQA-R (Ma et al., 2024) expanded the original test set by rephrasing each question 25 times via human-machine collaboration, scaling from 9,129 to 211,572 questions. It introduced a vocabulary of 465 words (5× larger than MUSIC-AVQA) and categorized questions into head (common) and tail (rare) samples to assess out-of-distribution performance. The dataset preserved training/validation biases but enabled granular evaluation through three metrics: head accuracy, tail accuracy, and overall accuracy. Most recently, FortisAVQA (Ma

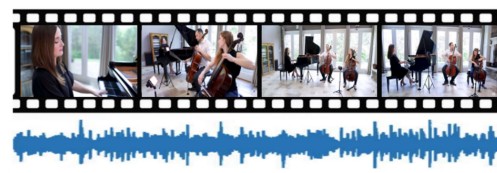 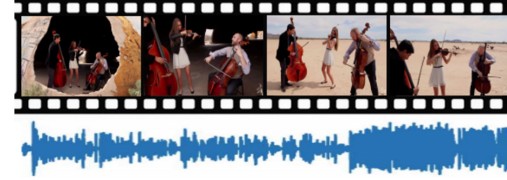

(a) Real performance.  (b) Synthetic video.

Figure 3: Representative samples from the MUSIC-AVQA dataset Li et al. (2022). (a) Question: Are there violin and cello sound? Answer: no. (b) Question: What is the instrument on the right of the piano? Answer: cello.

et al., 2025) replaced fixed threshold-based splitting with conformal-inspired optimization, dynamically adapting head-tail splits to ensure coverage and compactness across diverse distributions. Together, these datasets provide a rigorous framework for evaluating AVQA systems, from foundational benchmarks (MUSIC-AVQA) to bias-aware (v2.0) and robustness-focused (MUSIC-AVQA-R, FortisAVQA) iterations. Statistics are summarized in Table **??**, while Figure 4 illustrates real and synthetic samples.

Table 3: Question Statistics Comparison of MUSIC AVQA Datasets

| Dataset | #QA | #Type | Key Features |
|---|---|---|---|
| MUSIC-AVQA | 45.8K | 33 | 9 question types |
| MUSIC-AVQA2 | 54K | 33 | Balanced templates |
| MUSIC-AVQA-R | 211K | 33 | 25× rephrased; 465 vocab. |

### C.2 DATA PREPROCESSING

We applied a two-step preprocessing pipeline to both MUSIC-AVQA and Fortis-AVQA datasets: (1) audio-video separation, and (2) JSON reformatting.

**1. Audio-Video Separation.** For each video, we used `MoviePy` to extract the audio track and save it as a `.wav` file. The video (without audio) was saved as a new `.mp4` file. The process was parallelized with `multiprocessing.Pool`. Corrupted files were detected via exceptions and logged for exclusion.

```
for video in video_list:
    try:
        clip = VideoFileClip(video)
        clip.audio.write_audiofile(out_audio)
        clip.without_audio().write_videofile(out_video)
    except Exception as e:
        log_corrupted(video)
```

Listing 1: Audio-Video Separation (Simplified)

**2. JSON Reformatting.** Original JSON entries were converted to a unified EQUALS-compatible format. For each entry, placeholders in `question_content` (e.g., `<LR>`) were replaced using values in `templ_values`.

```
for entry in data:
    question = fill_template(entry["question_content"],
                             entry["templ_values"])
    new_entry = {
```

```
5        "video_path": get_video_path(entry["video_id"]),
6        "audio_path": get_audio_path(entry["video_id"]),
7        "image_path": "",
8        "text": "",
9        "answer": entry["answer"],
10       "question": question,
11       "task": entry["type"]
12   }
```

Listing 2: JSON Preprocessing (Simplified)

All transformed samples were saved into new JSON files with a standardized directory layout.

# D    TRAINING DETAILS

## D.1    TEXT, VIDEO AND AUDIO ENCODING

Text inputs are encoded using the CLIP-ViT-Large-Patch14 transformer, with the [CLS] token used as the global semantic representation to calculate pooling attention score, meanwhile also embedded with Llama3 to get the more fine-grained token-level representation for alignment operation. Video frames are sampled uniformly at 2 Hz, with each 10-frame clip processed into 256 spatial patches using a tubelet size of 1, resulting in a 1408-dimensional embedding space. For audio, we segment clips into 10-frame windows, each with 25 frequency bands and downsampled to 16k Hz. A dual-branch architecture using BEATs and Whisper v3 encodes the audio frames, and sinusoidal positional encoding is applied, resulting in 2048-dimensional embeddings.

## D.2    POOLING AND ALIGNMENT

We use a global pooling layer with a softmax activation, a pooling size of 512, and pooling temperature of 0.1 to control smoothness. An OT loss is applied to guide cross-modality alignment, with a loss weight coefficient of 0.3.

## D.3    BACKBONE AND MoE STRUCTURE

The core model is based on DeepSeek-MoE (deepseek-moe-16b-chat variant), employing a top-2 routing strategy. It includes 16 routed experts and 2 shared experts, with 4 experts selected per token. The hidden size and output size are 4096, the intermediate size is 8192, and the MoE-specific intermediate size is 1024. Expert scoring uses a softmax function with normalized top-$k$ probability. An auxiliary loss is applied with a coefficient of 0.001, and sequence-level auxiliary supervision is enabled. The activation function is SiLU.

## D.4    OPTIMIZATION AND TRAINING

Training is performed on $4\times$NVIDIA A100 GPUs using mixed-precision (fp16). The initial learning rate is set to $3 \times 10^{-5}$, with a 200-step warm-up starting from $1 \times 10^{-6}$, followed by cosine decay to a minimum of $3 \times 10^{-6}$. We use the Adam optimizer with a gradient accumulation step of 4. The training and evaluation batch size is 2. We fix the random seed to 42 across all libraries to ensure reproducibility.

The training leveraged DeepSpeed with gradient accumulation (4 steps, micro-batch size 2/GPU) and gradient clipping (max norm=3.0). FP16 mixed-precision training employed dynamic loss scaling (1-256 range, 1000-step window). Memory optimization used ZeRO stage 1 for optimizer state partitioning. The AdamW optimizer (lr=2×10, =(0.9,0.999), =1×10) incorporated decoupled weight decay (0.01).

# E    FRAMEWORK COMPONENTS

**Intervideo**    Wang et al. (2022) integrates generative and discriminative learning by employing Masked Autoencoders (MAE) for action understanding and video-language contrastive learning for

semantic alignment. Cross-model attention (CMA) modules further enhance representations during supervised training. This unified approach achieves superior action recognition, video-language alignment, and open-world task performance with efficient training.

**Whisper**  Radford et al. (2023) employs a Transformer-based encoder-decoder trained on vast web-scale audio-transcript pairs using weak supervision. Its multitask training framework enables a single model to handle speech recognition, translation, and speaker identification by representing tasks as token sequences. This design enhances generalization, robustness, and adaptability without fine-tuning on specific datasets.

**Beats**  Chen et al. (2022b) iteratively optimizes acoustic tokenizers and audio SSL models by converting audio signals into semantically rich discrete labels. Starting with a random-projection tokenizer clustering audio features, subsequent iterations refine the tokenizer via knowledge distillation from the SSL model. Using a Vision Transformer backbone and a Masked Audio Modeling task, BEATS enhances label prediction, achieving state-of-the-art audio classification with superior robustness and semantic alignment.

**DeepSeekMoE**  Dai et al. (2024) enhances expert specialization and reduces redundancy using fine-grained expert segmentation and shared expert isolation. This approach splits experts into smaller units with flexible activation and assigns some experts to capture common knowledge. It achieves significant performance gains, nearly matching dense models at 2B parameters and equaling DeepSeek 7B at 16B parameters with only 40% computation, while supporting strong scalability and adaptability.

## F  BASELINE MODELS

### F.1  TRADITIONAL DEEP LEARNING MODELS

**AVSD**  Alamri et al. (2019) employs a late-fusion architecture: visual features are extracted using a pretrained I3D model, audio features are obtained via a pretrained AENet, while both the dialog history and the current question are encoded using LSTM networks. These four modalities are then fused at the semantic level to form a unified representation, which is used to rank a set of candidate answers and select the most appropriate one. This architecture enables effective integration of multimodal context for generating accurate and contextually relevant responses in dynamic scenes.

**PSTP-Net**  Li et al. (2023) progressively identifies spatio-temporal regions relevant to the question through a series of modules. The Temporal Segment Selection Module (TSSM) locates key video segments, the Spatial Region Selection Module (SRSM) pinpoints important visual regions within these segments, and the Audio-guided Visual Attention Module (AVAM) performs fine-grained perception to associate audio with visual content. This step-by-step approach reduces redundancy and enhances both efficiency and accuracy.

**LAViT**  Yun et al. (2021) addresses spherical spatial and audio-visual reasoning in 360° videos by integrating visual, audio, and language modalities through separate unimodal encoders and a multi-modal encoder. It uses quaternion-based spatial representations for spherical geometry and includes an audio skewness prediction task. Visual features come from Faster R-CNN, audio features from a VGG-like CNN on stereo channels, and language features from BERT. Answers and grounding are predicted via a decoder.

**LAVisH**  Lin et al. (2023) adapts frozen Vision Transformers for audio-visual tasks by injecting trainable latent tokens that compress modality-specific information and enable efficient cross-modal fusion via attention bottlenecks. Its bidirectional design transfers information between audio and visual modalities, improving joint representation. It achieves strong performance on audio-visual tasks with far fewer tunable parameters.

### F.2 MULTIMODAL LARGE LANGUAGE MODELS

**MAVEN** Ma et al. (2025) employs a Multifaceted Cycle Collaborative Debiasing (MCCD) strategy to mitigate bias and improve generalization across in- and out-of-distribution scenarios. It extracts unimodal representations with modality-specific encoders, fuses them via a fine-tuned generative model, and captures unimodal biases through separate bias learners. The MCCD strategy amplifies distribution differences between unimodal and multimodal logits while maintaining bias consistency via cycle guidance, reducing reliance on harmful biases.

**VITA** Fu et al. (2024) builds on Mixtral 8×7B and extends its vocabulary through bilingual instruction tuning to enhance understanding of Chinese and English. It integrates visual and audio modalities into the language model using multimodal alignment and instruction tuning, introducing state tokens to differentiate input query types and enabling interaction without wake words. VITA employs a dual-model deployment architecture: one for response generation and another for continuous environmental monitoring, supporting audio-interruption-based interaction.

**VideoLLaMA 2** Cheng et al. (2024) uses a Spatial-Temporal Convolution connector and an Audio Branch to process visual and audio data separately before fusion in a fine-tuned large language model. The STC reduces tokens while preserving spatiotemporal order, and the Audio Branch employs BEATs for audio features and joint training. This design achieves state-of-the-art video and audio understanding.

**QWen2.5-VL** Bai et al. (2023) integrates a powerful visual encoder with the Qwen-7B language model and employs a three-stage training pipeline: pre-training on web-scale image-text pairs, multi-task pre-training with fine-grained data, and supervised fine-tuning for dialogue enhancement. It excels in multilingual conversations, multi-image interactions, and fine-grained visual understanding, outperforming existing vision-language models.

**GPT-4o** Hurst et al. (2024) employs a shared Transformer backbone to jointly model text, audio, and visual inputs in a common semantic space. Through multimodal instruction tuning, it achieves robust cross-modal alignment and task understanding, enabling coherent text generation, low-latency spoken dialogue, and visual reasoning like image interpretation and multimodal question answering. Its native multimodal training enhances generalization and inference efficiency across diverse real-world applications.

## G ADDITIONAL ANALYSIS OF QUESTION-GUIDED POOLING

### G.1 AVERAGE POOLING VS. QUESTION GUIDED POOLING

Tables 4 and 5 compare average pooling with our Question-Guided Pooling (QGP) under joint fusion. QGP consistently improves performance across both MUSIC-AVQA and FortisAVQA.

Table 4: Average Pooling vs.Question Guided Pooling (Joint Fusion) on AVQA tasks of the MUSIC-AVQA dataset.

| Method | Audio | | | Visual | | | Audio-Visual | | | | | | Avg |
|---|---|---|---|---|---|---|---|---|---|---|---|---|---|
| | Count | Comp | Avg | Count | Local | Avg | Exist | Count | Local | Comp | Temp | Avg | |
| AP(joint) | 85.97 | 81.44 | 85.31 | 83.31 | 89.88 | 85.10 | 87.38 | 79.57 | 85.49 | 81.85 | 84.61 | 85.53 | 85.33 |
| **QGP(joint)** | 88.46 | 82.95 | 86.43 | 83.39 | 91.09 | 88.28 | 91.11 | 82.00 | 86.47 | 82.95 | 86.37 | 85.25 | 86.36 |
| Δ (QGP–AP) | +2.49 | +1.51 | +1.12 | +0.08 | +1.21 | +3.18 | +3.73 | +2.43 | +0.98 | +1.10 | +1.76 | -0.28 | +1.03 |

### G.2 ATTENTIVE VISUALIZATION ANALYSIS

To verify the effectiveness of our proposed Question-Guided Pooling mechanism, we visualize the attention scores computed between the question semantics and video frames over time. As shown in Figure 4, we take a representative example where the question is "*Is the instrument on the left louder than the instrument on the right?*". Our model generates a temporal attention map based on

Table 5: Average Pooling vs. Question Guided Pooling (Joint Fusion) on FortisAVQA benchmark. (H: Head; T: Tail).

| Method | Audio QA | | | | Visual QA | | | | AVQA | | | | | | | | | | Avg |
|---|---|---|---|---|---|---|---|---|---|---|---|---|---|---|---|---|---|---|---|
| | Count | | Comp | | Count | | Local | | Exist | | Count | | Local | | Comp | | Temp | | |
| | H | T | H | T | H | T | H | T | H | T | H | T | H | T | H | T | H | T | |
| AP(joint) | 78.43 | 78.35 | 76.00 | 85.83 | 83.65 | 63.16 | 85.15 | 85.68 | 89.10 | 83.33 | 77.82 | 48.19 | 76.69 | 85.19 | 73.93 | 68.26 | 76.89 | 88.61 | 80.23 |
| **QGP(joint)** | 80.16 | 79.35 | 77.33 | 87.50 | 86.06 | 64.47 | 86.41 | 86.87 | 90.23 | 88.46 | 78.86 | 50.00 | 80.68 | 86.30 | 79.92 | 78.28 | 77.86 | 89.60 | 82.15 |
| $\Delta(QGP - AP)$ | +1.73 | +1.00 | +1.33 | +1.67 | +2.41 | +1.31 | +1.26 | +1.19 | +1.13 | +5.13 | +1.04 | +1.81 | +3.99 | +1.11 | +5.99 | +10.02 | +0.97 | +0.99 | +1.92 |

the semantic alignment between the question and the visual frames. Notably, the visual attention consistently emphasizes the spatial regions associated with "instrument", "left", and "right" across frames, aligning well with the core semantics of the question. The 4 attention weights, shown below each frame, further highlight the temporal consistency and discriminative capacity of learned attention. These results demonstrate the interpretability and efficacy of our Question-Guided Pooling strategy in capturing modality-question interactions for audiovisual reasoning.

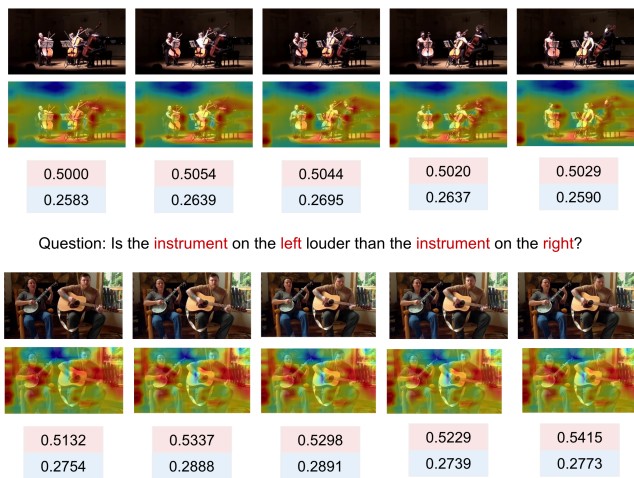

Figure 4: Attention visualization of Question-Guided Pooling. Red and blue boxes denote the highest and lowest attention scores at each frame respectively. Effective and interpretable guidance from question semantics is demonstrated by the meaningful attention distribution. Above: MUSIC-AVQA sample. Below: FortisAVQA sample.

## H  OTLOSS ABLATION ANALYSIS

We conducted ablation study with our best average-performance models to verify the effectiveness of OTLoss module. As shown in Table 6, the full OTLoss ($\lambda_{at} + \lambda_{vt} + \lambda_{av}$) achieves the best accuracy, with 86.17% on AVQA and 82.15% on FortisAVQA. Removing $\lambda_{av}$ leads to a drop of 2.37%/1.37%, while excluding $\lambda_{vt}$ and $\lambda_{at}$ results in a 1.83%/1.10% decrease, respectively. The baseline without OT losses performs worst (83.18%/79.90%). Similar trends are observed on FortisAVQA-Head and Tail subsets, where removing OT losses causes performance drops of 3.86% and 4.11%, respectively. These findings emphasize that the integration of cross-modal alignment through combined optimal losses leads to optimal performance, with the audio-text ($\lambda_{at}$) and video-text ($\lambda_{vt}$) losses proving to be particularly effective.

Table 6: Performance (%) across configurations with different OT loss components. Red numbers denote relative improvements over the *None* baseline.

| Configuration | AVQA | Fortis | Fortis$^H$ | Fortis$^T$ |
|---|---|---|---|---|
| $\lambda_{at} + \lambda_{vt} + \lambda_{av}$ | **86.17** (+2.99) | **82.15** (+2.25) | **82.76** (+4.07) | **82.25** (+6.99) |
| $\lambda_{at} + \lambda_{vt}$ | 84.34 (+1.16) | 81.05 (+1.15) | 81.56 (+2.87) | 80.09 (+4.83) |
| $\lambda_{av}$ | 83.80 (+0.62) | 80.78 (+0.88) | 80.48 (+1.79) | 78.14 (+2.88) |
| None | 83.18 | 79.90 | 78.69 | 75.26 |

## I  KERNEL AND STRIDE CONFIGURATIONS

We investigate the impact of different kernel and stride configurations under the *joint routing* fusion strategy. As shown in Figure 5, the best performance is achieved with an audio and visual kernel-stride setting of **[6,8]** and **[6,6,8]**, reaching an accuracy of 86.36% on MUSIC-AVQA, 82.15% on FortisAVQA and 82.25% on FortisAVQA-Tail test set respectively, except for FortisAVQA-Head set, the best accuracy 82.76% is achieved when kernel is set as **[8,10]** and stride as **[8,8,10]**. Interestingly, the results suggest that smaller kernel sizes do not always yield better performance. Instead, a comparatively larger but task-appropriate kernel and stride configuration strikes a better balance between spatial-temporal compression efficiency and model effectiveness, highlighting the importance of task-adaptive design rather than simply adopting minimal receptive fields.

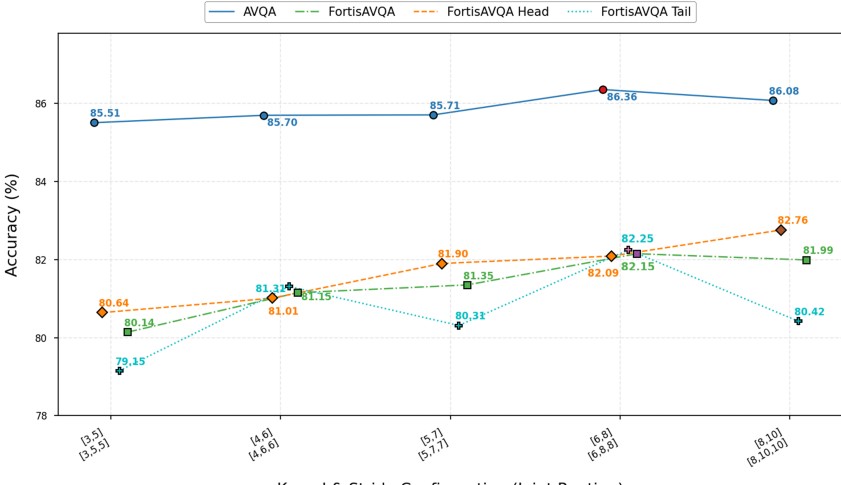

Figure 5: Accuracy comparison of different kernel and stride configurations under the per-modality routing fusion strategy. The best performance is achieved primarily with the [6,8], [6,6,8] configuration, except [8,10], [8,8,10] for FortisAVQA-Head test set, suggesting that in most case moderately-sized kernels and stides can better balance compression and representation capacity than smaller ones in most cases.

## J  QUALITATIVE EXAMPLES

We present qualitative comparisons between EQUALS and Qwen2.5-VL on MUSIC-AVQA. Each example shows the question, ground-truth answer, and model predictions. As illustrated in Figures 6–8, EQUALS consistently provides correct answers for audio-dependent queries, while Qwen2.5-VL often fails due to the lack of audio modality.

## A  QUALITATIVE EXAMPLES

We provide qualitative case studies comparing our EQUALS framework with Qwen2.5-VL. Each figure shows a representative sample from MUSIC-AVQA or FortisAVQA, with the corresponding

question, ground truth, and predictions. We provide qualitative case studies comparing EQUALS with Qwen2.5-VL. While Qwen2.5-VL performs reasonably well on purely visual queries, it fails to capture audio-dependent reasoning. In these cases, its outputs are either **incorrect** or based on **speculative guesses from visual cues only**, without leveraging the audio modality. In contrast, EQUALS consistently produces accurate predictions by explicitly incorporating audio features, demonstrating the necessity of cross-modal alignment for robust AVQA.

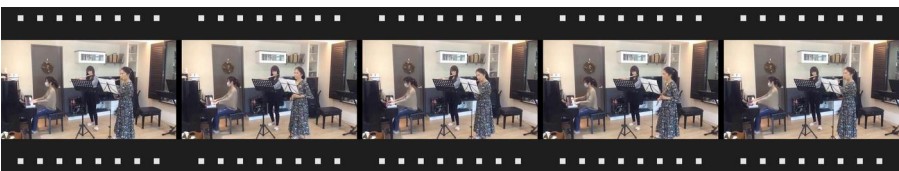

Figure 6: Sample eva00001188.mp4.

**Sample ID**: eva00001188.mp4

| | |
|---:|:---|
| **Question**: | Where is the lowest instrument? |
| **Ground Truth**: | left |
| **EQUALS**: | left |
| **Qwen2.5-VL**: | Predicts drum set at the **bottom** of the stage. |

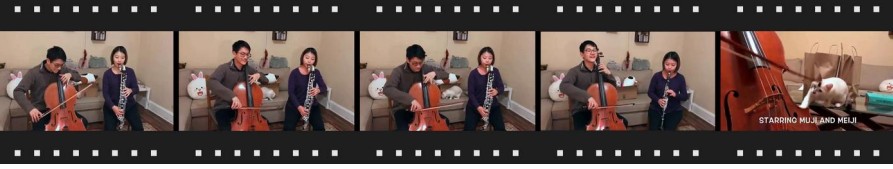

Figure 7: Sample 00004924.mp4.

**Sample ID**: 00004924.mp4

| | |
|---:|:---|
| **Question**: | How many instruments did not sound from beginning to end? |
| **Ground Truth**: | zero |
| **EQUALS**: | zero |
| **Qwen2.5-VL**: | In the video, there are three instruments: violin, cello, and piano. The piano does not play continuously, and thus is **predicted as the instrument that did not sound from beginning to end**. |

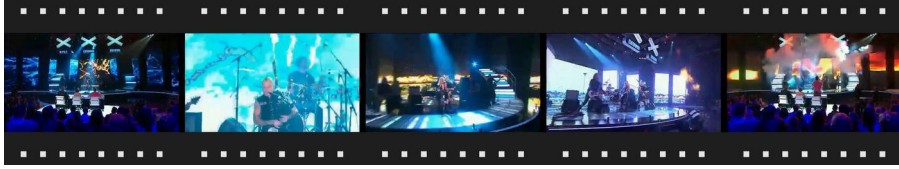

Figure 8: Sample 00004166.mp4.

**Sample ID**: 00004166.mp4

**Question**: Is this sound from the instrument in the video?
**Ground Truth**: yes
**EQUALS**: yes
**Qwen2.5-VL**: The video shows three individuals in a practice room. One plays piano, another a wind instrument, and a third sings. **The model does not confirm the audio source, stating it is not possible without sound**.

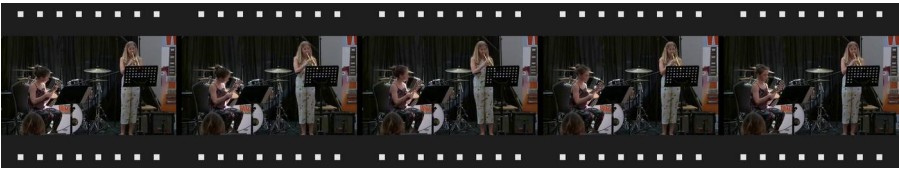

Figure 9: Sample 00007348.mp4.

**Sample ID**: 00007348.mp4

**Question**: What is the left instrument of the first sounding instrument?
**Ground Truth**: trumpet
**EQUALS**: trumpet
**Qwen2.5-VL**: The left instrument in the image is a trombone, which is being played by the musician on the left. The first sounding instrument, which is the one that starts playing, **appears to be the trombone**.

