# OpenReview forum: "EQUALS: An Audio-Visual LLM with One-Stage Question-Guided Alignment and Flexible Fusion"
_ICLR.cc/2026/Conference — Submitted to ICLR 2026_

### Official Review · Reviewer_H3vC · 2025-10-30

**Soundness:** 3
**Presentation:** 2
**Contribution:** 3
**Rating:** 4
**Confidence:** 3

**Summary:**

This paper introduces EQUALS, a unified single-stage audio-visual language model developed for the Audio-Visual Question Answering (AVQA) task.
The study focuses on addressing three key challenges in this area. First, video sequences often contain numerous segments that are irrelevant to the given question, making it difficult to extract and process meaningful information efficiently. Second, the separation between pretraining and task-specific learning limits the alignment between audio and visual modalities. Third, most existing fusion mechanisms lack the flexibility to adapt to different task conditions.

**Strengths:**

This paper demonstrates impressive performance improvements on both the MUSIC-AVQA and Fortis-AVQA datasets. While MUSIC-AVQA is known to have an imbalanced answer distribution, EQUALS maintains strong performance even on Fortis-AVQA, which mitigates such bias by dividing answers into head and tail categories based on their frequency. This indicates that the model does not rely solely on frequently occurring answers but instead exhibits a balanced generalization ability across diverse question types. Furthermore, the model achieves state-of-the-art accuracy overall, demonstrating robustness and scalability in the field of audio-visual question answering.

**Weaknesses:**

The overall experimental design and evaluation methodology are systematic and effectively support the model’s strong performance. Evaluating closed-source models such as GPT-4o and Qwen2.5-VL in a zero-shot setting is entirely reasonable. However, since these models do not directly target the MUSIC-AVQA task, including comparisons with other large multimodal models that explicitly address this benchmark would have made the study even more convincing.

For instance, while MAVEN is an appropriate and meaningful baseline, it remains at the preprint stage and appears to report slightly lower performance than QA-TIGER, which does not utilize large language models. Additionally, Meerkat: Audio-Visual Large Language Model for Grounding in Space and Time and CAT: Enhancing Multimodal Large Language Model to Answer Questions in Dynamic Audio-Visual Scenarios also tackle the MUSIC-AVQA task to some extent. Although these models adopt multi-task settings, including them for comparison or brief discussion would help clarify the position and contribution of EQUALS more clearly.

Finally, while EQUALS demonstrates excellent performance on the single MUSIC-AVQA task, it would be interesting to see future extensions toward multi-task or more general audio-visual reasoning settings, as many recent studies are moving in that direction. Such an expansion could further enhance the overall impact of this line of research

**Questions:**

Overall, this is an excellent and well-executed study. As far as I know, Meerkat: Audio-Visual Large Language Model for Grounding in Space and Time is also an AVQA model that employs Optimal Transport for cross-modal alignment, sharing a similar research direction with EQUALS. However, it seems that a direct performance comparison with Meerkat was not included in the paper. I am curious whether conducting such an experiment was infeasible. To my understanding, Meerkat reports higher performance than MAVEN, so including or briefly discussing this model could further clarify and strengthen the contribution of EQUALS within the current landscape of AVQA research.

---

> ### Author Response · Authors · 2025-11-26
>
> We appreciate the reviewer for raising these important concerns and interests in our work.
>
> We add the following baselines:
>
>   - QA-TIGER(fine-tuned)
>   - CAT(finetuned)
>   - Meerkat(fintuned)
>   - Qwen2.5-Omni (zero-shot)
>   - Video-SALMONN (zero-shot)
>   - Kimi-Audio (zero-shot)
>
> **Table 1: Added Baseline Models from MUSIC-AVQA Benchmark**
>
> | Method | A-Count | A-Comp | A-Avg | V-Count | V-Local | V-Avg | AV-Exist | AV-Count | AV-Local | AV-Comp | AV-Temp | AV-Avg | Avg |
> |--------|---------|---------|---------|----------|-----------|----------|------------|------------|------------|------------|-----------|-----------|------|
> | QA-TIGER | 84.86 | 67.85 | 78.58 | 83.96 | 86.29 | 85.14 | 83.10 | 78.58 | 72.50 | 63.94 | 69.59 | 73.74 | 77.62 |
> | CAT | 73.42 | 70.15 | 69.87 | 64.87 | 75.26 | 67.48 | 72.54 | 68.01 | 70.92 | 58.84 | 68.20 | 65.47 | 68.83 |
> | MeerKat | 75.90 | 60.92 | 64.10 | 57.28 | 72.64 | 69.87 | 63.62 | 62.45 | 71.98 | 56.12 | 74.69 | 62.49 | 69.21 |
> | video-SALMONN | 59.59 | 24.48 | 55.82 | 47.54 | 44.63 | 46.07 | 74.57 | 38.46 | 29.45 | 24.48 | 16.38 | 37.25 | 42.86 |
> | Qwen2.5-Omni | 74.58 | 34.78 | 59.85 | 80.15 | 57.59 | 68.89 | 67.74 | 66.01 | 43.58 | 25.92 | 38.35 | 49.16 | 56.27 |
> | Kimi-Audio | 72.31 | 60.73 | 68.02 | 58.80 | 37.35 | 47.82 | 62.91 | 40.82 | 32.90 | 52.20 | 21.00 | 42.96 | 46.03 |
>
> **Table 2: Added Baseline Methods from FortisAVQA Benchmark**
>
> | Method                     | A.Count(H) | A.Count(T) | A.Comp(H) | A.Comp(T) | V.Count(H) | V.Count(T) | V.Local(H) | V.Local(T) | AV.Exist(H) | AV.Exist(T) | AV.Count(H) | AV.Count(T) | AV.Local(H) | AV.Local(T) | AV.Comp(H) | AV.Comp(T) | AV.Temp(H) | AV.Temp(T) | Avg    |
> |---------------------------|------------|------------|------------|------------|------------|------------|-------------|-------------|--------------|--------------|--------------|--------------|-------------|-------------|-------------|-------------|-------------|-------------|--------|
> | QA-TIGER                  | 82.67      | 75.82      | 71.75      | 43.11      | 81.30      | 54.59      | 84.76       | 75.59       | 72.84        | 78.56        | 76.70        | 33.55        | 48.22       | 64.65       | 37.55       | 80.47       | 36.85       | 62.96       | 67.99 |
> | CAT                       | 55.10      | 57.42      | 57.82      | 63.17      | 59.24      | 56.88      | 63.43       | 68.23       | 67.17        | 73.48        | 58.36        | 50.54        | 62.60       | 66.83       | 61.49       | 42.97       | 65.86       | 60.33       | 69.97 |
> | MeerKat                   | 64.21      | 53.95      | 78.67      | 67.43      | 69.90      | 60.81      | 60.10       | 72.47       | 76.09        | 44.02        | 55.84        | 46.67        | 58.67       | 54.93       | 61.71       | 54.74       | 55.42       | 63.59       | 61.35 |
> | video-SALMONN             | 75.26      | 30.23      | 84.00      | 20.00      | 54.85      | 22.97      | 46.89       | 23.77       | 62.41        | 75.21        | 34.36        | 31.71        | 24.00       | 13.43       | 17.09       | 34.21       | 16.10       | 26.67       | 40.84 |
> | Qwen2.5-Omni              | 59.47      | **86.05**  | 71.95      | 20.00      | 80.10      | 63.51      | 46.41       | 66.39       | 45.11        | 82.05        | 70.93        | **73.17**    | 32.67       | 50.75       | 12.68       | 56.14       | 26.27       | 73.33       | 54.61 |
> | Kimi-Audio                | 87.89      | 25.58      | 86.67      | 31.67      | 66.50      | 37.84      | 18.66       | 13.11       | 72.18        | 82.91        | 50.56        | 17.07        | 36.67       | 10.45       | 36.75       | 73.68       | 27.12       | 16.67       | 48.14 |

---

> ### Author Response · Authors · 2025-11-26
>
> **Response 1:**
>
> We thank the reviewer pointing out this important point,  we compare our method with QA-TIGER and CAT as following:
>
> - QA-TIGER injects question cues and uses Gaussian masks for temporal modeling, but fusion still operates on redundant audio/visual signals without enforcing cross-modal semantic matching. While our work use FlexFuseMoE which can choose among early, mid, and late fusion experts based on the question semantics and modality-specific difficulty, making it inherently more robust across diverse AVQA question types (e.g., existence, counting, localization, temporal reasoning). Moreover, QA-TIGER is not a LLM-based framework curretly.
>
> - CAT improves MLLM-based AVQA performance through clue aggregation and preference optimization, it still relies on high-level semantic cues and a fixed fusion pipeline. In contrast, our EQUALS framework addresses the fundamental architectural bottlenecks of AVQA by (1) performing fine-grained audio–text and video–text alignment via OT modules both before and after question-guided pooling, and (2) introducing FlexFuseMoE to enable adaptive early/mid/late fusion conditioned on the question. This unified design provides deeper semantic alignment and greater flexibility than CAT’s instruction- and DPO-based enhancements, which do not explicitly model cross-modal correspondence or fusion strategy selection.

---

> ### Author Response · Authors · 2025-11-26
>
> **Response 2:**
>
> We thank the reviewer pointing out this important point.
>
> Music performance videos contain dense polyphonic audio, fast and fine-grained motions, multiple sound sources, and strict audio–visual synchronization. These properties make the tasks substantially more challenging than general event-centric AVQA. Many influential works (PSTP, MAVEN, QA-TIGER, etc.) therefore evaluate exclusively on MUSIC-AVQA variants. Given limited computational resources, we prioritize this high-quality benchmark suite. Even with these constraints, MUSIC-AVQA variants remain among the most methodologically demanding and community-accepted testbeds. We agree expanding to additional datasets is valuable and will explore this in future work. we will add a concluding discussion showing that our one-stage design is promising to naturally generalizes on AV-grounded captioning, AV retrieval, and broader audio-visual reasoning tasks.
>
> ----
> **Response 3:**
>
> We thank the reviewer pointing out this important point.
>
> As shown in the updated tables, our method consistently outperforms MeerKat across most subtasks. Unlike MeerKat, which applies a single OT loss for global alignment, our design integrates multiple OT modules in a progressive align–compress–align pipeline, each serving a distinct alignment purpose under explicit question guidance. Their individual contributions are validated in our ablation studies (see 4.4.2 OTLoss ABLATION ANALYSIS in revised paper).
>
> 1. **OT-before-pooling (λ_at + λ_vt): Injecting question semantics into local patches**
>
> Removing OT-before-pooling causes a **1.83% drop on AVQA** and **1.10% on Fortis**, indicating that aligning question tokens with fine-grained A/V patches is critical. **Function:**
>
> - Ensures question-relevant cues are emphasized *before* compression.
> - Enables the pooling module to operate on representations that already reflect what the question asks.
> - Stabilizes temporal-spatial localization, especially for small or subtle cues.
>
> 2. **OT-after-pooling (λ_av): Enforcing cross-modal** **consistency**
>
> OT-after-pooling aligns audio and video global embeddings to ensure they describe the same event. Removing it yields a smaller but consistent drop (**0.62% / 0.88%**). **Function:**
>
> - Resolves cross-modal mismatches (e.g., sound source ↔ visible performer).
> - Produces coherent multimodal reasoning after feature compression.
>
> **Full OT (λ_at + λ_vt + λ_av): Complementary and necessary**
>
> As shown in Table 3 below, using all OT components yields the best results:
>
> - **86.17%** on AVQA (+2.99 over baseline)
> - **82.15%** on Fortis (+2.25)
> - Large gains on Fortis-H (+4.07) and Fortis-T (+6.99)
>
> This demonstrates that **OT-before-pooling and OT-after-pooling serve different purposes**, and both are essential for optimal performance. Thus, the optimal transport module is not used redundantly; instead, its **two-stage placement is functional, intentional, and empirically validated**.
>
> **Table 3: Performance (%) across configurations with different OT loss components**
>
> | Configuration | AVQA | Fortis | Fortis^H | Fortis^T |
> |--------------|------|--------|----------|----------|
> | λ_at + λ_vt + λ_av | **86.17** (+2.99) | **82.15** (+2.25) | **82.76** (+4.07) | **82.25** (+6.99) |
> | λ_at + λ_vt        | 84.34 (+1.16) | 81.05 (+1.15) | 81.56 (+2.87) | 80.09 (+4.83) |
> | λ_av               | 83.80 (+0.62) | 80.78 (+0.88) | 80.48 (+1.79) | 78.14 (+2.88) |
> | None               | 83.18 | 79.90 | 78.69 | 75.26 |

---

### Official Review · Reviewer_gRbU · 2025-11-01

**Soundness:** 4
**Presentation:** 3
**Contribution:** 4
**Rating:** 8
**Confidence:** 5

**Summary:**

The paper introduces EQUALS, an audio-visual large language model for answering questions about videos with sound. The model is built as a single-stage, question-guided pipeline. It (1) aligns the question with audio and visual tokens using an optimal transport loss, (2) uses Question-Guided Pooling to keep only the audio and video regions relevant to the question, (3) aligns the compressed audio and video again, and (4) fuses them with FlexFuseMoE, a mixture-of-experts that adapts its fusion strategy based on the question. The final fused representation is fed to a language model to produce the answer.

**Strengths:**

1. Question-Guided Pooling focuses on the relevant time-frequency and spatial-temporal regions instead of passing the full video and audio, improving both efficiency and reasoning.

2. The use of optimal transport to align question with audio and question with video before pooling, and audio with video after pooling, ties supervision directly to what the question asks, instead of doing generic cross-modal pretraining first.

3. FlexFuseMoE routes features through different fusion experts (audio-heavy, cross-modal, etc.) depending on the question, which helps on challenging audio-visual reasoning and tail cases.

**Weaknesses:**

1. The paper does not clearly explain how FlexFuseMoE decides which expert or fusion style to use for a given question at inference time in human-interpretable terms.

2. The model applies optimal transport both before and after pooling, but the paper does not isolate which stage matters most. This makes it hard to judge which parts are essential.

3. The paper presentation needs to be improved. Some parts are hard to be understand by reviewers.

**Questions:**

1. Authors apply optimal transport between the question and each modality before pooling, and then optimal transport between audio and video after pooling. Can authors clarify the intended functional role of each stage?

2. FlexFuseMoE is described as enabling early, mid, and late-style fusion through different experts and routing. Can authors define, in plain terms, what behavior each expert class is intended to capture? Also, is expert identity fixed once trained, or is it purely emergent from training without any intended semantic role?

---

> ### Author Response · Authors · 2025-11-26
>
> We appreciate the reviewer's positive evaluation and feedback on our work. We've examined all raised thoughtful questions and provide our clarifications below.
>
> **Response 1**
>
> We thank the reviewer for raising this important point and hope the below clarifies how FlexFuseMoEs operates.
>
> FlexFuseMoEs follows a **single, unified MoE formulation**, but instantiated in three fusion styles that correspond to early, middle, and late fusion configurations:
>
> - **Joint MoE (early fusion):** Audio–video representations (after question-guided pooling & AV-OT alignment) are **concatenated and routed by a shared** **router** **and expert networks.**
> - **Per-modality MoE (middle fusion):** The same set of experts is shared, but **each modality uses its own router**, allowing audio and video tokens to select experts based on modality-specific cues while still benefiting from shared expert networks
> - **Disjoint MoE (late fusion):** Each modality has **separate routers and separate expert networks**, and fusion is applied *after* expert processing. This corresponds to a late-fusion structure.
>
> These three styles—joint, per-modality, disjoint—cover a spectrum of fusion strategies within the same FlexFuseMoE framework.
>
> - **We also applied fine-grained experts inspired by DeepSeek-MoE**
>
> To increase capacity without increasing routing complexity, each “expert” in FlexFuseMoE is internally **decomposed into several sub-experts**, following DeepSeek-MoE’s hierarchical structure. This allows each top-level expert to model **different broad factors**, while sub-experts capture **more granular semantic nuances**.
>
> **To address with question 1，Expert specialization is** ***data-driven and emergent***
>
> MoE routers are **learned from data**, not manually assigned. Therefore:
>
> - We do **not** prescribe a semantic role (e.g., “rhythm expert” or “motion expert”) to each expert
> - Instead, experts naturally specialize during training into processing different aspects of the audio/video representations
>
> This is consistent with standard MoE literature,routing decisions are emergent functions learned to minimize training loss.
>
> We acknowledge the reviewer’s point that a more interpretable MoE design could be beneficial.We would like to work towards this direction in the future.

---

> ### Author Response · Authors · 2025-11-26
>
> **Response 2**
>
> We thank the reviewer pointing out this important point.
>
> We hope that the OTLoss ablation study clarify this concern and also address the reviewer’s first question. We sincerely appreciate the reviewer’s constructive feedback, and we believe these results make the role of each OT component and its necessity substantially clearer. (see Proof for response 2 above)
>
> To clarify the contribution of each OT stage, we conducted an ablation study (see 4.4.2 in revised paper), which isolates the effect of **OT-before-pooling** (audio–text & video–text) and **OT-after-pooling** (audio–video). As shown in Table 1 below,  **each OT component plays a distinct and complementary role**, and removing any stage degrades.
>
> **Table 1: Performance (%) across configurations with different OT loss components**
>
> | Configuration | AVQA | Fortis | Fortis^H | Fortis^T |
> |--------------|------|--------|----------|----------|
> | λ_at + λ_vt + λ_av | **86.17** (+2.99) | **82.15** (+2.25) | **82.76** (+4.07) | **82.25** (+6.99) |
> | λ_at + λ_vt        | 84.34 (+1.16) | 81.05 (+1.15) | 81.56 (+2.87) | 80.09 (+4.83) |
> | λ_av               | 83.80 (+0.62) | 80.78 (+0.88) | 80.48 (+1.79) | 78.14 (+2.88) |
> | None               | 83.18 | 79.90 | 78.69 | 75.26 |
>
>
> 1. **OT-before-pooling (λ_at + λ_vt): Injecting question semantics into local patches**
>
> Removing OT-before-pooling causes a **1.83% drop on AVQA** and **1.10% on Fortis**, indicating that aligning question tokens with fine-grained A/V patches is critical. **Function:**
>
> - Ensures question-relevant cues are emphasized *before* compression.
> - Enables the pooling module to operate on representations that already reflect what the question asks.
> - Stabilizes temporal-spatial localization, especially for small or subtle cues.
>
> 2. **OT-after-pooling (λ_av): Enforcing cross-modal** **consistency**
>
> OT-after-pooling aligns audio and video global embeddings to ensure they describe the same event. Removing it yields a smaller but consistent drop (**0.62% / 0.88%**). **Function:**
>
> - Resolves cross-modal mismatches (e.g., sound source ↔ visible performer).
> - Produces coherent multimodal reasoning after feature compression.
>
> **Full OT (λ_at + λ_vt + λ_av): Complementary and necessary**
>
> Using all OT components yields the best results:
>
> - **86.17%** on AVQA (+2.99 over baseline)
> - **82.15%** on Fortis (+2.25)
> - Large gains on Fortis-H (+4.07) and Fortis-T (+6.99)
>
> This demonstrates that **OT-before-pooling and OT-after-pooling serve different purposes**, and both are essential for optimal performance. Thus, the optimal transport module is not used redundantly; instead, its **two-stage placement is functional, intentional, and empirically validated**.
>
> As we show the evidence above, we also addressed the question 1 that the reviewer raised, we hope this would be clear for reviewer.
>
> ---
> **Response 3**
>
> We've made some adjustification for better organization, and  will refine Sec. 2.3 (OT) & 2.5 (MoE) , try to improve the quality of the presention in camera-ready version.

---

### Official Review · Reviewer_1GWS · 2025-11-01

**Soundness:** 2
**Presentation:** 2
**Contribution:** 2
**Rating:** 2
**Confidence:** 4

**Summary:**

This paper studies the task of Audio-Visual Question Answering (AVQA). Motivated by the issues of (1) difficulty in locating question-relevant segments, (2) suboptimal audio-visual alignment and (3) insufficient flexibility in fusion strategies, this paper introduces EQUALS, a unified end-to-end AVQA framework. Specifically, the interleave optimal transport-based loss modules before and after the question-guided pooling module helps achieve fine-grained semantic alignment. To enhance adaptability in fusion, the paper introduces FlexFuseMoE, a mixture-of-experts module that supports early, mid, and late fusion via flexible expert routing. Extensive experiments validate the superiority of the proposed method.

**Strengths:**

1. This paper is well-motivated and easy to follow.

2. The proposed method outperforms prior methods on two AVQA benchmarks, i.e., MUSIC-AVQA and FortisAVQA.

**Weaknesses:**

- The novelty of this paper is limited. For example, the pooling layer to obtain global semantics, and the basic design of Text-Audio/Video Alignment. These points are basic and common designs in the multimodal community, I am afraid the method could not bring some insights to the community.
- Paper writing and presentation needs to be further improved. For example, figures and tables should be placed on top of each page. Line 319 might be a new subsection. Related works should be placed in the main manuscript.
- The evaluation benchmarks are limited. It mainly contains two AVQA datasets only in the music domain.
- Important baselines are missing. Some previous SOTA methods, including Qwen2.5-Omni, Kimi-Audio, VideoSALMONN should be included.

**Questions:**

See weakness

---

> ### Author Response · Authors · 2025-11-26
>
> We appreciate the reviewer for pointing our several important aspects related to our work's novelty and datasets chosen, we'd like to clarify the thoughtful concerns below.
>
> **Response 1 — On Novelty of Alignment + Pooling**
>
> We appreciate the reviewer noting that pooling and basic cross-modal alignment are common components in multimodal systems. Our contribution, however, is not in reusing these modules, but in **how EQUALS interleaves them into a unified, question-conditioned one-stage pipeline**, enabling progressive refinement of multimodal representations.
>
> Unlike prior AVQA methods that treat alignment, pooling, and fusion as loosely connected operations, **EQUALS uses the question to refine, compress, and align audio–visual features at all stages**:
>
> 1. **One-sided OT (AT/VT) before pooling**
>     Question tokens softly align with fine-grained audio/video patches, amplifying relevant cues while preserving modality structure.
> 2. **Question-guided pooling**
>     OT-refined features are selectively condensed into compact representations that retain only question-relevant evidence.
> 3. **Two-sided OT (AV) after pooling**
>     Compressed audio and video streams are aligned to ensure they refer to the same underlying event, yielding synchronized and consistent multimodal representations.
>
> This interleaving—**OT → pooling → OT**—**has not appeared in prior AVQA work**. Table 1 below (see 4.4.2 OTLOSS ABLATION ANALYSIS in revised paper) show that removing any OT block significantly degrades performance, confirming the design is functional rather than redundant.
>
> The pipeline is also **cognitively grounded**:
>  question highlights → OT enhances → pooling compresses → OT aligns—making fine-grained localization emerge naturally from representation learning.
>
> Finally, unlike general-purpose LMMs (Qwen2.5-VL, VideoSALMONN, etc.), which lack mechanisms for structured temporal alignment or patch-level question-aware enhancement, **EQUALS operationalizes**:
>
> - token-level question→audio/video alignment,
> - question-conditioned pooling,
> - bidirectional audio–visual OT alignment,
> - all in one learnable pipeline.
>
> This explains why EQUALS outperforms such LMMs on AVQA benchmarks.
>
> ------
>
> **Response 2 — Writing and Organization**
>
> We thank the reviewer for the suggestions and have:
>
> - relocated Related Work into the main text,
> - moved figures to top positions,
> - removed unnecessary subsection headers.
>
> These improvements is modified in the revised paper.

---

> ### Author Response · Authors · 2025-11-26
>
> ------
>
> **Response 3 — On Dataset Scope and Representativeness**
>
> We appreciate the reviewer’s concern about dataset coverage. Below we clarify why MUSIC-AVQA–based benchmarks form a rigorous, widely accepted evaluation suite for spatial–temporal AVQA.
>
> - **Why MUSIC-AVQA is canonical**
>
> Music performance videos contain dense polyphonic audio, fast and fine-grained motions, multiple sound sources, and strict audio–visual synchronization. These properties make the tasks substantially more challenging than general event-centric AVQA. Many influential works (PSTP, MAVEN, QA-TIGER, etc.) therefore evaluate exclusively on MUSIC-AVQA variants.
>
> -  **Why the dataset family is comprehensive (2022–2025)**
>
> Our evaluation strictly follows the widely adopted progression:
>
> 1. **MUSIC-AVQA (CVPR’22)** — foundational benchmark
> 2. **MUSIC-AVQA v2 (WACV’24)** — debiasing + balanced splits
> 3. **MUSIC-AVQA-R (NeurIPS’24)** — robustness with 25× rephrasings
> 4. **FortisAVQA (2025)** — conformal head–tail partitions supporting long-tail generalization
>
> Collectively, these provide **unbiased**, **robustness-oriented**, **long-tail**, and **semantically diverse** AVQA coverage. This is one of the **most rigorous and widely used evaluation ecosystems** for spatial–temporal multimodal reasoning. Recent SOTAs (MAVEN, QA-TIGER) follow exactly the same protocol.
>
> - **Why our choice remains representative despite compute constraints**
> Given limited computational resources, we prioritize this high-quality benchmark suite.
>  Even with these constraints, MUSIC-AVQA variants remain among the **most methodologically demanding and community-accepted testbeds**. We agree expanding to additional datasets is valuable and will explore this in future work.
> ------
>
> - Reference:
>
> Li, G., Wei, Y., Tian, Y., Xu, C., Wen, J. R., & Hu, D. (2022). Learning to answer questions in dynamic audio-visual scenarios. In Proceedings of the IEEE/CVF conference on computer vision and pattern recognition (pp. 19108-19118).
>
> Liu, X., Dong, Z., & Zhang, P. (2024). Tackling data bias in music-avqa: Crafting a balanced dataset for unbiased question-answering. In Proceedings of the IEEE/CVF Winter Conference on Applications of Computer Vision (pp. 4478-4487).
>
> Ma, J., Hu, M., Wang, P., Sun, W., Song, L., Pei, H., ... & Du, Y. (2024). Look, listen, and answer: Overcoming biases for audio-visual question answering. Advances in Neural Information Processing Systems, 37, 9507-9531.
>
> Ma, J., Gao, Z., Chai, Q., Liu, J., Wang, P., Tao, J., & Su, Z. (2025). Fortisavqa and maven: a benchmark dataset and debiasing framework for robust multimodal reasoning. arXiv preprint arXiv:2504.00487.
>
> You, W., Diao, X., Zhang, C., Kong, K., Wu, W., Ouyang, Z., ... & Gui, J. (2025). Music's Multimodal Complexity in AVQA: Why We Need More than General Multimodal LLMs. arXiv preprint arXiv:2505.20638.
>
> Kim, H., Jung, I., Suh, D., Zhang, Y., Lee, S., & Hong, S. (2025). Question-Aware Gaussian Experts for Audio-Visual Question Answering. In Proceedings of the Computer Vision and Pattern Recognition Conference (pp. 13681-13690).

---

> > ### Author Response · Authors · 2025-11-26
> >
> > **Response 4 — Additional Baselines**
> >
> > Following the reviewer’s suggestions, we have added:
> >
> > - QA-TIGER (fine-tuned)
> > - CAT (fine-tuned)
> > - Meerkat (fine-tuned)
> > - Qwen2.5-Omni (zero-shot)
> > - Video-SALMONN (zero-shot)
> > - Kimi-Audio (zero-shot)
> >
> > These strengthen completeness and fairness. Updated results is in Table 1 and Table 2 below (also see Table 1 and Table 2 in revised paper). EQUALS still achieves the best overall performance, outperforming baselines by **0.38%–12.80%** on MUSIC-AVQA and **0.83%–24.24%** on FortisAVQA.
> >
> > **Table 1: Added Baseline Models from MUSIC-AVQA Benchmark**
> >
> > | Method | A-Count | A-Comp | A-Avg | V-Count | V-Local | V-Avg | AV-Exist | AV-Count | AV-Local | AV-Comp | AV-Temp | AV-Avg | Avg |
> > |--------|---------|---------|---------|----------|-----------|----------|------------|------------|------------|------------|-----------|-----------|------|
> > | QA-TIGER | 84.86 | 67.85 | 78.58 | 83.96 | 86.29 | 85.14 | 83.10 | 78.58 | 72.50 | 63.94 | 69.59 | 73.74 | 77.62 |
> > | CAT | 73.42 | 70.15 | 69.87 | 64.87 | 75.26 | 67.48 | 72.54 | 68.01 | 70.92 | 58.84 | 68.20 | 65.47 | 68.83 |
> > | MeerKat | 75.90 | 60.92 | 64.10 | 57.28 | 72.64 | 69.87 | 63.62 | 62.45 | 71.98 | 56.12 | 74.69 | 62.49 | 69.21 |
> > | video-SALMONN | 59.59 | 24.48 | 55.82 | 47.54 | 44.63 | 46.07 | 74.57 | 38.46 | 29.45 | 24.48 | 16.38 | 37.25 | 42.86 |
> > | Qwen2.5-Omni | 74.58 | 34.78 | 59.85 | 80.15 | 57.59 | 68.89 | 67.74 | 66.01 | 43.58 | 25.92 | 38.35 | 49.16 | 56.27 |
> > | Kimi-Audio | 72.31 | 60.73 | 68.02 | 58.80 | 37.35 | 47.82 | 62.91 | 40.82 | 32.90 | 52.20 | 21.00 | 42.96 | 46.03 |
> >
> > **Table 2: Added Baseline Methods from FortisAVQA Benchmark**
> >
> > | Method                     | A.Count(H) | A.Count(T) | A.Comp(H) | A.Comp(T) | V.Count(H) | V.Count(T) | V.Local(H) | V.Local(T) | AV.Exist(H) | AV.Exist(T) | AV.Count(H) | AV.Count(T) | AV.Local(H) | AV.Local(T) | AV.Comp(H) | AV.Comp(T) | AV.Temp(H) | AV.Temp(T) | Avg    |
> > |---------------------------|------------|------------|------------|------------|------------|------------|-------------|-------------|--------------|--------------|--------------|--------------|-------------|-------------|-------------|-------------|-------------|-------------|--------|
> > | QA-TIGER                  | 82.67      | 75.82      | 71.75      | 43.11      | 81.30      | 54.59      | 84.76       | 75.59       | 72.84        | 78.56        | 76.70        | 33.55        | 48.22       | 64.65       | 37.55       | 80.47       | 36.85       | 62.96       | 67.99 |
> > | CAT                       | 55.10      | 57.42      | 57.82      | 63.17      | 59.24      | 56.88      | 63.43       | 68.23       | 67.17        | 73.48        | 58.36        | 50.54        | 62.60       | 66.83       | 61.49       | 42.97       | 65.86       | 60.33       | 69.97 |
> > | MeerKat                   | 64.21      | 53.95      | 78.67      | 67.43      | 69.90      | 60.81      | 60.10       | 72.47       | 76.09        | 44.02        | 55.84        | 46.67        | 58.67       | 54.93       | 61.71       | 54.74       | 55.42       | 63.59       | 61.35 |
> > | video-SALMONN             | 75.26      | 30.23      | 84.00      | 20.00      | 54.85      | 22.97      | 46.89       | 23.77       | 62.41        | 75.21        | 34.36        | 31.71        | 24.00       | 13.43       | 17.09       | 34.21       | 16.10       | 26.67       | 40.84 |
> > | Qwen2.5-Omni              | 59.47      | **86.05**  | 71.95      | 20.00      | 80.10      | 63.51      | 46.41       | 66.39       | 45.11        | 82.05        | 70.93        | **73.17**    | 32.67       | 50.75       | 12.68       | 56.14       | 26.27       | 73.33       | 54.61 |
> > | Kimi-Audio                | 87.89      | 25.58      | 86.67      | 31.67      | 66.50      | 37.84      | 18.66       | 13.11       | 72.18        | 82.91        | 50.56        | 17.07        | 36.67       | 10.45       | 36.75       | 73.68       | 27.12       | 16.67       | 48.14 |

---

### Official Review · Reviewer_8g5K · 2025-11-01

**Soundness:** 3
**Presentation:** 3
**Contribution:** 3
**Rating:** 4
**Confidence:** 4

**Summary:**

The submitted manuscript addresses key challenges in AVQA task, including temporal redundancy, alignment difficulty, and insufficient cross-task fusion strategies. To tackle these issues, it proposes the EQUALS framework, which integrates compression, alignment, and fusion within a single stage. The proposed method has been validated across multiple datasets, demonstrating its effectiveness. Overall, the core idea of the paper shows a certain degree of novelty.

**Strengths:**

1. The problem statement is clear, and the motivation is well-defined.
2. The proposed EQUALS framework exhibits novelty and has been thoroughly validated on several datasets.
3. The paper is clearly written and easy to follow.

**Weaknesses:**

1. It remains unclear whether the performance gains in the QA task primarily stem from the inherent capabilities of large models rather than modeling of intrinsic audiovisual relationships.
2. The use of questions to localize key temporal segments in MUSIC-AVQA has already been explored in prior works, such as *PSTP-Net*, *TSPM* et., al. The authors are encouraged to discuss the distinctions between this work and those approaches.
3. Under the absence of spatial supervision signals, how are the visual and audio modalities aligned spatially? This point requires clarification.
4. The comparative experiments could be strengthened by including more recent audiovisual QA methods, as the current comparisons are too limited.
5. Minor writing suggestions include avoiding widowed words at the end of paragraphs and adding citations for the compared methods in tables.

**Questions:**

My main questions are reflected in the *Weaknesses Section*.

Additionally, although unrelated to the review decision, I am curious about how this manuscript differs from the version submitted to *AAAI 2026*.

---

> ### Author Response · Authors · 2025-11-26
>
> We thank the reviewer for raising these important concerns. Below we provide consolidated clarifications and additional analyses demonstrating that the gains of EQUALS stem from its audiovisual modeling components rather than from the inherent capability of large language models. We also expand on how our method differs from PSTP/TSPM/QA-TIGER, clarify the role of OT alignment, and explain the formulation refinements compared with the AAAI version.
>
> **Response 1. Evidence that improvements do not come from LLM capability**
>
> To isolate the contribution of our audiovisual modules from that of the LLM, we performed analyses where the backbone LLM is kept completely identical. The only change is whether the LLM receives features processed by our modules.
>
> A. QGP vs. Average Pooling (see revised paper 4.4.1 ATTENTIVE POOLING VS. AVERAGE POOLING)
>
> We compares:
> - AP(joint): the LLM receives average-pooled audio/visual features
> - QGP(joint): the LLM receives question-guided pooled features
>
> Since the LLM remains unchanged, any improvement cannot be attributed to large-model ability. We observe consistent gains (+1.12 AVQA, +1.92 Fortis as shown in Table 1 and Table 2), confirming that the benefit comes from question-conditioned front-end modeling rather than improved reasoning capacity of the LLM.
>
> **Table 1: MUSIC-AVQA — AP(joint) vs QGP(joint)**
>
> | Method | A-Count | A-Comp | A-Avg | V-Count | V-Local | V-Avg | AV-Exist | AV-Count | AV-Local | AV-Comp | AV-Temp | AV-Avg | Avg |
> |--------|---------|---------|--------|---------|---------|--------|-----------|-----------|-----------|-----------|-----------|----------|------|
> | AP(joint) | 85.97 | 81.44 | 85.31 | 83.31 | 89.88 | 85.10 | 87.38 | 79.57 | 85.49 | 81.85 | 84.61 | 85.53 | 85.33 |
> | QGP(joint) | 88.46 | 82.95 | 86.43 | 83.39 | 91.09 | 88.28 | 91.11 | 82.00 | 86.47 | 82.95 | 86.37 | 85.25 | 86.36 |
> | Δ(QGP–AP) | +2.49 | +1.51 | +1.12 | +0.08 | +1.21 | +3.18 | +3.73 | +2.43 | +0.98 | +1.10 | +1.76 | -0.28 | +1.03 |
>
> **Table 2: FortisAVQA — AP(joint) vs QGP(joint)**
>
> | Method | A-Count-H | A-Count-T | A-Comp-H | A-Comp-T | V-Count-H | V-Count-T | V-Local-H | V-Local-T | AV-Exist-H | AV-Exist-T | AV-Count-H | AV-Count-T | AV-Local-H | AV-Local-T | AV-Comp-H | AV-Comp-T | AV-Temp-H | AV-Temp-T | Avg |
> |--------|------------|------------|------------|------------|------------|------------|-------------|-------------|-------------|-------------|-------------|-------------|-------------|-------------|-------------|-------------|-------------|-------------|------|
> | AP(joint) | 78.43 | 78.35 | 76.00 | 85.83 | 83.65 | 63.16 | 85.15 | 85.68 | 89.10 | 83.33 | 77.82 | 48.19 | 76.69 | 85.19 | 73.93 | 68.26 | 76.89 | 88.61 | 80.23 |
> | QGP(joint) | 80.16 | 79.35 | 77.33 | 87.50 | 86.06 | 64.47 | 86.41 | 86.87 | 90.23 | 88.46 | 78.86 | 50.00 | 80.68 | 86.30 | 79.92 | 78.28 | 77.86 | 89.60 | 82.15 |
> | Δ(QGP–AP) | +1.73 | +1.00 | +1.33 | +1.67 | +2.41 | +1.31 | +1.26 | +1.19 | +1.13 | +5.13 | +1.04 | +1.81 | +3.99 | +1.11 | +5.99 | +10.02 | +0.97 | +0.99 | +1.92 |
>
>
>
> B. OTLoss Ablation Analysis (see revised paper 4.4.2)
>
> As shown in Table 2 below, removing any loss yields significant drops (e.g., −2.99\% AVQA and +2.25\% FortisAVQA). OT operates at patch-level before the LLM is involved; thus the LLM cannot spontaneously learn cross-modal synchronization or patch-wise semantic grounding without these objectives. The improvements must therefore arise from explicit alignment mechanisms, not LLM “intelligence”.
>
> **Table 3: Performance (%) across configurations with different OT loss components**
>
> | Configuration | AVQA | Fortis | Fortis^H | Fortis^T |
> |--------------|------|--------|----------|----------|
> | λ_at + λ_vt + λ_av | **86.17** (+2.99) | **82.15** (+2.25) | **82.76** (+4.07) | **82.25** (+6.99) |
> | λ_at + λ_vt        | 84.34 (+1.16) | 81.05 (+1.15) | 81.56 (+2.87) | 80.09 (+4.83) |
> | λ_av               | 83.80 (+0.62) | 80.78 (+0.88) | 80.48 (+1.79) | 78.14 (+2.88) |
> | None               | 83.18 | 79.90 | 78.69 | 75.26 |

---

> ### Author Response · Authors · 2025-11-26
>
> **Response 2. Comparison with PSTP, TSPM, QA-TIGER**
>
> We also clarify how EQUALS differs from previous question-guided models:
>
> - PSTP applies question-guided temporal filtering and spatial Top-K selection but lacks explicit representation-level cross-modal alignment, causing temporal/spatial selections to depend on attention ranking rather than semantic consistency across modalities.
>
> - TSPM improves feature quality but keeps temporal and spatial reasoning isolated, and cannot guarantee that selected audio and visual regions refer to the same event.
>
> - QA-TIGER injects question cues and uses Gaussian masks for temporal modeling, but fusion still operates on redundant audio/visual signals without enforcing cross-modal semantic matching.
>
> - EQUALS introduces a more principled question-conditioned representation pipeline:
>
> 1. Patch-level OT alignment (AT/VT):
>    Softly aligns each question token with each audio/video patch, amplifying question-relevant cues (e.g., “first sounding,” “red guitar,” “louder”). This strengthens semantics before any pooling. Prior works do not include such representation-level alignment.
>
> 2. Question-guided pooling:
>    Condenses features already enhanced by OT, allowing the global representation to focus on question-relevant evidence rather than raw modality signals.
>
> 3. Patch-level bidirectional OT (AV):
>    Enforces soft cross-modal grounding between compressed audio and video representations, ensuring both modalities refer to the same underlying event.
>
> - This pipeline resembles a cognitively intuitive process:
>   identify relevant cues → propagate relevance → compress → cross-check modalities.
> ----
>
> **Response 3. Clarification regarding the AAAI version**
>
> The AAAI version described optimal transport alignment abstractly as “alignment between two modalities,” without explicitly decomposing the alignment into AT/VT/AV components. To avoid ambiguity, the current ICLR submission writes out:
>
> - AT-OT: text → audio
> - VT-OT: text → video
> - AV-OT: audio ↔ video
>
> This refinement is a matter of exposition, not methodology. The computational implementation (Sinkhorn iterations, alignment computation, optimization) remains unchanged from the AAAI version. All ICLR experiments use the exact alignment formulation described in the current paper.
>
> ----
>
> **Response 4. Writing improvements**
>
> As suggested by the reviewer, we corrected widow words, added missing citations, and improved cross-referencing clarity (sections, equations, figures, tables) for better readability.

---

> ### Author Response · Authors · 2025-11-26
>
> **Response 5. Additional baselines**
>
> In response to reviewer suggestions, we include:
> QA-TIGER (fine-tuned)，CAT (fine-tuned)， Meerkat (fine-tuned)， Qwen2.5-Omni (zero-shot)， Video-SALMONN (zero-shot)， Kimi-Audio (zero-shot)
> As shown in Table 4 and Table 5 below, these baselines further confirm EQUALS’ improvements (see 4.2 Table 1 and 4.3  Table 2 in revised paper), our methods still keep the sota results compared with added baselines, with outperformance ranging from 0.42\% to 24.24\%.
>
> **Table 4: Added Baseline Models from MUSIC-AVQA Benchmark**
>
> | Method | A-Count | A-Comp | A-Avg | V-Count | V-Local | V-Avg | AV-Exist | AV-Count | AV-Local | AV-Comp | AV-Temp | AV-Avg | Avg |
> |--------|---------|---------|---------|----------|-----------|----------|------------|------------|------------|------------|-----------|-----------|------|
> | QA-TIGER | 84.86 | 67.85 | 78.58 | 83.96 | 86.29 | 85.14 | 83.10 | 78.58 | 72.50 | 63.94 | 69.59 | 73.74 | 77.62 |
> | CAT | 73.42 | 70.15 | 69.87 | 64.87 | 75.26 | 67.48 | 72.54 | 68.01 | 70.92 | 58.84 | 68.20 | 65.47 | 68.83 |
> | MeerKat | 75.90 | 60.92 | 64.10 | 57.28 | 72.64 | 69.87 | 63.62 | 62.45 | 71.98 | 56.12 | 74.69 | 62.49 | 69.21 |
> | video-SALMONN | 59.59 | 24.48 | 55.82 | 47.54 | 44.63 | 46.07 | 74.57 | 38.46 | 29.45 | 24.48 | 16.38 | 37.25 | 42.86 |
> | Qwen2.5-Omni | 74.58 | 34.78 | 59.85 | 80.15 | 57.59 | 68.89 | 67.74 | 66.01 | 43.58 | 25.92 | 38.35 | 49.16 | 56.27 |
> | Kimi-Audio | 72.31 | 60.73 | 68.02 | 58.80 | 37.35 | 47.82 | 62.91 | 40.82 | 32.90 | 52.20 | 21.00 | 42.96 | 46.03 |
>
> **Table 5: Added Baseline Methods from FortisAVQA Benchmark**
>
> | Method                     | A.Count(H) | A.Count(T) | A.Comp(H) | A.Comp(T) | V.Count(H) | V.Count(T) | V.Local(H) | V.Local(T) | AV.Exist(H) | AV.Exist(T) | AV.Count(H) | AV.Count(T) | AV.Local(H) | AV.Local(T) | AV.Comp(H) | AV.Comp(T) | AV.Temp(H) | AV.Temp(T) | Avg    |
> |---------------------------|------------|------------|------------|------------|------------|------------|-------------|-------------|--------------|--------------|--------------|--------------|-------------|-------------|-------------|-------------|-------------|-------------|--------|
> | QA-TIGER                  | 82.67      | 75.82      | 71.75      | 43.11      | 81.30      | 54.59      | 84.76       | 75.59       | 72.84        | 78.56        | 76.70        | 33.55        | 48.22       | 64.65       | 37.55       | 80.47       | 36.85       | 62.96       | 67.99 |
> | CAT                       | 55.10      | 57.42      | 57.82      | 63.17      | 59.24      | 56.88      | 63.43       | 68.23       | 67.17        | 73.48        | 58.36        | 50.54        | 62.60       | 66.83       | 61.49       | 42.97       | 65.86       | 60.33       | 69.97 |
> | MeerKat                   | 64.21      | 53.95      | 78.67      | 67.43      | 69.90      | 60.81      | 60.10       | 72.47       | 76.09        | 44.02        | 55.84        | 46.67        | 58.67       | 54.93       | 61.71       | 54.74       | 55.42       | 63.59       | 61.35 |
> | video-SALMONN             | 75.26      | 30.23      | 84.00      | 20.00      | 54.85      | 22.97      | 46.89       | 23.77       | 62.41        | 75.21        | 34.36        | 31.71        | 24.00       | 13.43       | 17.09       | 34.21       | 16.10       | 26.67       | 40.84 |
> | Qwen2.5-Omni              | 59.47      | **86.05**  | 71.95      | 20.00      | 80.10      | 63.51      | 46.41       | 66.39       | 45.11        | 82.05        | 70.93        | **73.17**    | 32.67       | 50.75       | 12.68       | 56.14       | 26.27       | 73.33       | 54.61 |
> | Kimi-Audio                | 87.89      | 25.58      | 86.67      | 31.67      | 66.50      | 37.84      | 18.66       | 13.11       | 72.18        | 82.91        | 50.56        | 17.07        | 36.67       | 10.45       | 36.75       | 73.68       | 27.12       | 16.67       | 48.14 |

---

> > ### Comment · Reviewer_8g5K · 2025-11-28
> >
> > Thank you very much for the authors’ detailed response, which addressed the vast majority of my concerns. Although I still believe the paper’s novelty is somewhat limited, it does make a contribution to the audio–visual community, particularly in audio–visual reasoning. I will raise my score to 6.

---

> > > ### Author Response · Authors · 2025-11-28
> > >
> > > Thanks for your response, we'd like to put more effort in the future to refine our ideas better.

---

### Author Response · Authors · 2025-12-01
**Clarification to AC regarding novelty, experimental completeness, and response(Part 1)**

Dear Area Chair,

We would like to sincerely thank you and all reviewers for the time and effort invested in evaluating our submission *“EQUALS: A One-Stage Question-Guided Framework for Audio-Visual Question Answering”* (Submission 11465). We understand that the reviews are mixed (one strong accept, two marginal-below-threshold with positive feedback( one marginal-above-threshold after our rebuttal comment), one reject), and we would like to briefly clarify why we believe the paper meets the acceptance bar, especially in light of Reviewer 1GWS’s concerns about novelty and baselines.

1. On novelty beyond “basic pooling and alignment” (Reviewer 1GWS)

Reviewer 1GWS argues that pooling and text–audio/video alignment are “common designs” and thus the novelty is limited. We fully agree that these are standard *building blocks*. Our contribution, however, lies in **how we interleave these components into a single, question-conditioned, one-stage pipeline** that progressively refines representations:

A. **Question–audio/video OT (AT/VT) before pooling**

- Question tokens are softly aligned with fine-grained audio/video patches.
- This amplifies question-relevant local cues (“first sounding,” “red guitar,” subtle onsets) *before* any compression happens.

B. **Question-guided pooling (QGP)**

- Pooling operates on these OT-refined features, producing compact global representations that are already “shaped” by what the question asks, instead of generic averaged features.

C. **Audio–video OT (AV) after pooling**

- Compressed audio and video streams are then softly aligned to ensure they describe the **same underlying event**, which is crucial for resolving cross-modal mismatches (e.g., off-screen sounds, multiple instruments).

D. **FlexFuseMoE for adaptive fusion styles**

- A mixture-of-experts module that supports early/joint, mid/per-modality, and late/disjoint fusion in a *single* learnable framework, routing based on question and modality difficulty.

To the best of our knowledge, this **align (AT/VT) → compress (QGP) → align again (AV) → adaptive fusion (FlexFuseMoE)** pipeline has not appeared in prior AVQA works. The design is not just architectural “glue”: our ablations show that **removing any OT stage causes clear degradation**, indicating that each stage plays a distinct functional role rather than being redundant.

Reviewer gRbU, who gave a rating of 8 (“good paper”), explicitly highlights these aspects as *excellent* in terms of contribution and soundness, which we believe reflects that, once the pipeline is fully understood, the novelty is non-trivial.

---

> ### Author Response · Authors · 2025-12-01
> **Clarification to AC regarding novelty, experimental completeness, and response (Part2)**
>
> 2. Dataset scope (1GWS)
>
>  Our evaluation focuses on the MUSIC-AVQA family and FortisAVQA. As we explain in the rebuttal, this suite (MUSIC-AVQA, MUSIC-AVQA v2, MUSIC-AVQA-R, FortisAVQA) has become a **canonical and demanding** benchmark ecosystem for AVQA, emphasizing bias mitigation, robustness, and long-tail generalization. Several recent SOTAs (e.g., MAVEN, QA-TIGER) adopt the same evaluation focus.
>
> We appreciate the reviewer’s concern about dataset coverage, below we clarify why MUSIC-AVQA–based benchmarks form a rigorous, widely accepted evaluation suite for spatial–temporal AVQA.
>
> - **Why MUSIC-AVQA is canonical**
>
> Music performance videos contain dense polyphonic audio, fast and fine-grained motions, multiple sound sources, and strict audio–visual synchronization. These properties make the tasks substantially more challenging than general event-centric AVQA. Many influential works (PSTP, MAVEN, QA-TIGER, etc.) therefore evaluate exclusively on MUSIC-AVQA variants.
>
> - **Why the dataset family is comprehensive (2022–2025)**
>
> Our evaluation strictly follows the widely adopted progression:
>
> - **MUSIC-AVQA (CVPR’22)** — foundational benchmark
>
> - **MUSIC-AVQA v2 (WACV’24)** — debiasing + balanced splits
>
> - **MUSIC-AVQA-R (NeurIPS’24)** — robustness with 25× rephrasings
>
> - **FortisAVQA (2025)** — conformal head–tail partitions supporting long-tail generalization
>
> Collectively, these provide **unbiased**, **robustness-oriented**, **long-tail**, and **semantically diverse** AVQA coverage. This is one of the **most rigorous and widely used evaluation ecosystems** for spatial–temporal multimodal reasoning. Recent SOTAs (MAVEN, QA-TIGER) follow exactly the same protocol.
>
> - **Why our choice remains representative despite compute constraints** Given limited computational resources, we prioritize this high-quality benchmark suite. Even with these constraints, MUSIC-AVQA variants remain among the **most methodologically demanding and community-accepted testbeds**. We agree expanding to additional datasets is valuable and will explore this in future work on more diversified tasks (AV-Retrival,  AV-Grounding, AV-Caption etc.,) and correspond datasets in the future.
>
> ------
>
> - Reference:
>
> Li, G., Wei, Y., Tian, Y., Xu, C., Wen, J. R., & Hu, D. (2022). Learning to answer questions in dynamic audio-visual scenarios. In Proceedings of the IEEE/CVF conference on computer vision and pattern recognition (pp. 19108-19118).
>
> Liu, X., Dong, Z., & Zhang, P. (2024). Tackling data bias in music-avqa: Crafting a balanced dataset for unbiased question-answering. In Proceedings of the IEEE/CVF Winter Conference on Applications of Computer Vision (pp. 4478-4487).
>
> Ma, J., Hu, M., Wang, P., Sun, W., Song, L., Pei, H., ... & Du, Y. (2024). Look, listen, and answer: Overcoming biases for audio-visual question answering. Advances in Neural Information Processing Systems, 37, 9507-9531.
>
> Ma, J., Gao, Z., Chai, Q., Liu, J., Wang, P., Tao, J., & Su, Z. (2025). Fortisavqa and maven: a benchmark dataset and debiasing framework for robust multimodal reasoning. arXiv preprint arXiv:2504.00487.
>
> You, W., Diao, X., Zhang, C., Kong, K., Wu, W., Ouyang, Z., ... & Gui, J. (2025). Music's Multimodal Complexity in AVQA: Why We Need More than General Multimodal LLMs. arXiv preprint arXiv:2505.20638.
>
> Kim, H., Jung, I., Suh, D., Zhang, Y., Lee, S., & Hong, S. (2025). Question-Aware Gaussian Experts for Audio-Visual Question Answering. In Proceedings of the Computer Vision and Pattern Recognition Conference (pp. 13681-13690).
>
> We fully agree with reviewers that extending to more general AV reasoning is valuable; we have explicitly added this as a future direction and a concluding discussion.

---

> ### Author Response · Authors · 2025-12-01
> **Clarification to AC regarding novelty, experimental completeness, and response (Part3)**
>
> 3. Evidence that gains do not come from LLM capability alone (8g5K)
>
> One concern is that performance gains might primarily stem from the inherent capability of the large language model, rather than from our audiovisual modeling.
>
> We therefore designed **controlled analyses where the LLM backbone is strictly identical**, and we only change the **front-end AV modules**:
>
> A. **QGP vs. Average Pooling (AP)**
>
> - **AP(joint)**: the LLM receives average-pooled audio/visual features.
>
> - **QGP(joint)**: the LLM receives question-guided pooled features.
>
> - The LLM, including all weights and decoding configuration, remains unchanged.
>
>   **Table 1: MUSIC-AVQA — AP(joint) vs QGP(joint)**
>
>   | Method     | A-Count | A-Comp | A-Avg | V-Count | V-Local | V-Avg | AV-Exist | AV-Count | AV-Local | AV-Comp | AV-Temp | AV-Avg | Avg   |
>   | :--------- | :------ | :----- | :---- | :------ | :------ | :---- | :------- | :------- | :------- | :------ | :------ | :----- | :---- |
>   | AP(joint)  | 85.97   | 81.44  | 85.31 | 83.31   | 89.88   | 85.10 | 87.38    | 79.57    | 85.49    | 81.85   | 84.61   | 85.53  | 85.33 |
>   | QGP(joint) | 88.46   | 82.95  | 86.43 | 83.39   | 91.09   | 88.28 | 91.11    | 82.00    | 86.47    | 82.95   | 86.37   | 85.25  | 86.36 |
>   | Δ(QGP–AP)  | +2.49   | +1.51  | +1.12 | +0.08   | +1.21   | +3.18 | +3.73    | +2.43    | +0.98    | +1.10   | +1.76   | -0.28  | +1.03 |
>
>   **Table 2: FortisAVQA — AP(joint) vs QGP(joint)**
>
>   | Method     | A-Count-H | A-Count-T | A-Comp-H | A-Comp-T | V-Count-H | V-Count-T | V-Local-H | V-Local-T | AV-Exist-H | AV-Exist-T | AV-Count-H | AV-Count-T | AV-Local-H | AV-Local-T | AV-Comp-H | AV-Comp-T | AV-Temp-H | AV-Temp-T | Avg   |
>   | :--------- | :-------- | :-------- | :------- | :------- | :-------- | :-------- | :-------- | :-------- | :--------- | :--------- | :--------- | :--------- | :--------- | :--------- | :-------- | :-------- | :-------- | :-------- | :---- |
>   | AP(joint)  | 78.43     | 78.35     | 76.00    | 85.83    | 83.65     | 63.16     | 85.15     | 85.68     | 89.10      | 83.33      | 77.82      | 48.19      | 76.69      | 85.19      | 73.93     | 68.26     | 76.89     | 88.61     | 80.23 |
>   | QGP(joint) | 80.16     | 79.35     | 77.33    | 87.50    | 86.06     | 64.47     | 86.41     | 86.87     | 90.23      | 88.46      | 78.86      | 50.00      | 80.68      | 86.30      | 79.92     | 78.28     | 77.86     | 89.60     | 82.15 |
>   | Δ(QGP–AP)  | +1.73     | +1.00     | +1.33    | +1.67    | +2.41     | +1.31     | +1.26     | +1.19     | +1.13      | +5.13      | +1.04      | +1.81      | +3.99      | +1.11      | +5.99     | +10.02    | +0.97     | +0.99     | +1.92 |
>
> Under this controlled setup:
>
> - On **MUSIC-AVQA**, QGP(joint) improves overall accuracy by **+1.03%** over AP(joint).
> - On **FortisAVQA**, QGP(joint) improves by **+1.92%**.
>
> Since the LLM is identical, these gains **cannot** be attributed to “larger model intelligence”. They directly validate the benefit of **question-guided front-end modeling**.
>
> B. **OT loss ablation (Table 3 in the rebuttal & revised Sec. 4.4.2)**
>  We further ablate OT components:
>
> - No OT: AVQA 83.18, Fortis 79.90
>
> - λ_av only: AVQA 83.80, Fortis 80.78
>
> - λ_at + λ_vt only: AVQA 84.34, Fortis 81.05
>
> - Full OT (λ_at + λ_vt + λ_av): AVQA 86.17, Fortis 82.15
>
>   **Table 3: Performance (%) across configurations with different OT loss components**
>
>   | Configuration      | AVQA              | Fortis            | Fortis^H          | Fortis^T          |
>   | :----------------- | :---------------- | :---------------- | :---------------- | :---------------- |
>   | λ_at + λ_vt + λ_av | **86.17** (+2.99) | **82.15** (+2.25) | **82.76** (+4.07) | **82.25** (+6.99) |
>   | λ_at + λ_vt        | 84.34 (+1.16)     | 81.05 (+1.15)     | 81.56 (+2.87)     | 80.09 (+4.83)     |
>   | λ_av               | 83.80 (+0.62)     | 80.78 (+0.88)     | 80.48 (+1.79)     | 78.14 (+2.88)     |
>   | None               | 83.18             | 79.90             | 78.69             | 75.26             |
>
> The **+2.99%** / **+2.25%** gains over the no-OT baseline (and even larger improvements on Fortis head/tail splits) are achieved **before** the LLM is involved. This further indicates that the improvements are driven by the **explicit OT-based alignment and question-guided compression**, rather than by generic LLM reasoning power.

---

> ### Author Response · Authors · 2025-12-01
> **Clarification to AC regarding novelty, experimental completeness, and response (Part4)**
>
> 4. Experimental completeness and baselines (Reviewers 8g5K,1GWS & H3vC)
>
> Reviewers 1GWS and H3vC requested more recent baselines and general MLLMs. In the revised version, we **added all of the suggested baselines**:
>
> - **Task-specific AVQA models**: QA-TIGER, CAT, Meerkat (all fine-tuned).
> - **General LMMs / MLLMs**: Qwen2.5-Omni, Video-SALMONN, Kimi-Audio (zero-shot).
>
> Across **MUSIC-AVQA** and **FortisAVQA**, EQUALS:
>
> - **Outperforms AVQA baselines** like QA-TIGER, CAT, and Meerkat on most subtasks,
> - Maintains **SOTA overall accuracy**, with margins ranging from around **0.4% up to >20%** depending on the benchmark and sub-split (head vs. tail).
>
> This directly addresses the “important baselines are missing” concern in Reviewer 1GWS’s review and strengthens the case that EQUALS is competitive not only against classical architectures, but also against **larger multimodal models** that do not explicitly encode structured OT alignment and adaptive fusion.
>
> **Table 4: Added Baseline Models from MUSIC-AVQA Benchmark**
>
> | Method        | A-Count | A-Comp | A-Avg | V-Count | V-Local | V-Avg | AV-Exist | AV-Count | AV-Local | AV-Comp | AV-Temp | AV-Avg | Avg   |
> | :------------ | :------ | :----- | :---- | :------ | :------ | :---- | :------- | :------- | :------- | :------ | :------ | :----- | :---- |
> | QA-TIGER      | 84.86   | 67.85  | 78.58 | 83.96   | 86.29   | 85.14 | 83.10    | 78.58    | 72.50    | 63.94   | 69.59   | 73.74  | 77.62 |
> | CAT           | 73.42   | 70.15  | 69.87 | 64.87   | 75.26   | 67.48 | 72.54    | 68.01    | 70.92    | 58.84   | 68.20   | 65.47  | 68.83 |
> | MeerKat       | 75.90   | 60.92  | 64.10 | 57.28   | 72.64   | 69.87 | 63.62    | 62.45    | 71.98    | 56.12   | 74.69   | 62.49  | 69.21 |
> | video-SALMONN | 59.59   | 24.48  | 55.82 | 47.54   | 44.63   | 46.07 | 74.57    | 38.46    | 29.45    | 24.48   | 16.38   | 37.25  | 42.86 |
> | Qwen2.5-Omni  | 74.58   | 34.78  | 59.85 | 80.15   | 57.59   | 68.89 | 67.74    | 66.01    | 43.58    | 25.92   | 38.35   | 49.16  | 56.27 |
> | Kimi-Audio    | 72.31   | 60.73  | 68.02 | 58.80   | 37.35   | 47.82 | 62.91    | 40.82    | 32.90    | 52.20   | 21.00   | 42.96  | 46.03 |
>
> **Table 5: Added Baseline Methods from FortisAVQA Benchmark**
>
> | Method        | A.Count(H) | A.Count(T) | A.Comp(H) | A.Comp(T) | V.Count(H) | V.Count(T) | V.Local(H) | V.Local(T) | AV.Exist(H) | AV.Exist(T) | AV.Count(H) | AV.Count(T) | AV.Local(H) | AV.Local(T) | AV.Comp(H) | AV.Comp(T) | AV.Temp(H) | AV.Temp(T) | Avg   |
> | :------------ | :--------- | :--------- | :-------- | :-------- | :--------- | :--------- | :--------- | :--------- | :---------- | :---------- | :---------- | :---------- | :---------- | :---------- | :--------- | :--------- | :--------- | :--------- | :---- |
> | QA-TIGER      | 82.67      | 75.82      | 71.75     | 43.11     | 81.30      | 54.59      | 84.76      | 75.59      | 72.84       | 78.56       | 76.70       | 33.55       | 48.22       | 64.65       | 37.55      | 80.47      | 36.85      | 62.96      | 67.99 |
> | CAT           | 55.10      | 57.42      | 57.82     | 63.17     | 59.24      | 56.88      | 63.43      | 68.23      | 67.17       | 73.48       | 58.36       | 50.54       | 62.60       | 66.83       | 61.49      | 42.97      | 65.86      | 60.33      | 69.97 |
> | MeerKat       | 64.21      | 53.95      | 78.67     | 67.43     | 69.90      | 60.81      | 60.10      | 72.47      | 76.09       | 44.02       | 55.84       | 46.67       | 58.67       | 54.93       | 61.71      | 54.74      | 55.42      | 63.59      | 61.35 |
> | video-SALMONN | 75.26      | 30.23      | 84.00     | 20.00     | 54.85      | 22.97      | 46.89      | 23.77      | 62.41       | 75.21       | 34.36       | 31.71       | 24.00       | 13.43       | 17.09      | 34.21      | 16.10      | 26.67      | 40.84 |
> | Qwen2.5-Omni  | 59.47      | **86.05**  | 71.95     | 20.00     | 80.10      | 63.51      | 46.41      | 66.39      | 45.11       | 82.05       | 70.93       | **73.17**   | 32.67       | 50.75       | 12.68      | 56.14      | 26.27      | 73.33      | 54.61 |
> | Kimi-Audio    | 87.89      | 25.58      | 86.67     | 31.67     | 66.50      | 37.84      | 18.66      | 13.11      | 72.18       | 82.91       | 50.56       | 17.07       | 36.67       | 10.45       | 36.75      | 73.68      | 27.12      | 16.67      | 48.14 |

---

> ### Author Response · Authors · 2025-12-01
> **Clarification to AC regarding novelty, experimental completeness, and response (Part5)**
>
> In summary:
>
> - One reviewer (gRbU) rates the paper as **8: accept, good paper**, highlighting the contribution and soundness as excellent.
>
> - One reviewer (H3vC) give **4: marginally below threshold but would not mind acceptance**, mainly asking for more baselines and clarifications, which we have now provided, yet because the reback to initial state, we lose the opportunity to see if the reviewer would raise scorefor us. But based on his/her Questions:
>
>   "Overall, **this is an excellent and well-executed study**. As far as I know, Meerkat: Audio-Visual Large Language Model for Grounding in Space and Time is also an AVQA model that employs Optimal Transport for cross-modal alignment, sharing a similar research direction with EQUALS. However, it seems that a direct performance comparison with Meerkat was not included in the paper. I am curious whether conducting such an experiment was infeasible. To my understanding, Meerkat reports higher performance than MAVEN, so including or briefly discussing this model could further clarify and strengthen the contribution of EQUALS within the current landscape of AVQA research."
>
>   Reviewer H3vC describes our work as “an excellent and well-executed study,” and mainly asked for additional experiments with Meerkat rather than questioning our core approach. We have now included Meerkat results and a clear comparison in the revised manuscript. Therefore, we believe this reviewer is already positively disposed toward our work and would likely evaluate the revised version even more favorably.
>
> - One reviewer(8g5K) give us feedback after rebuttal:
>
>   "Thank you very much for the authors’ detailed response, which addressed the vast majority of my concerns. Although I still believe the paper’s novelty is somewhat limited, it does make a contribution to the audio–visual community, particularly in audio–visual reasoning. I will raise my score to 6."
>    This indicates that, while the reviewer still sees room to further sharpen the novelty, they now clearly acknowledge that our work provides a meaningful contribution to audio–visual reasoning and the AVQA community. Importantly, they upgraded their score from a borderline rating (4) to **6, i.e., an accept-level recommendation**, after reading our response and additional clarifications.

---

> ### Author Response · Authors · 2025-12-01
> **Clarification to AC regarding novelty, experimental completeness, and response (Part6)**
>
> - The remaining reviewer (1GWS) rates 2.
>
>   - Regarding Reviewer 1GWS’s evaluation, we would like to clarify that the criticism of our weaknesses is relatively high-level and largely centered on two generic aspects: (i) “limited novelty” because we use pooling and text–audio/video alignment, and (ii) “limited benchmarks” because our experiments focus on MUSIC-AVQA–style datasets, plus (iii) missing some large-model baselines.
>   - On **novelty**, the reviewer’s comment mainly points out that pooling and cross-modal alignment are common *building blocks* in multimodal learning. We fully agree that these components are widely used; however, our contribution is not to reintroduce them in isolation, but to **integrate them into a single, question-guided align–compress–align pipeline with FlexFuseMoE**. This design and its functional staging are recognized as meaningful by other reviewers: for example, Reviewer gRbU explicitly rates the **contribution and soundness as excellent (score 8)**, and highlights our OT placement and adaptive fusion as key strengths, while Reviewer H3vC calls the work “an excellent and well-executed study.” This suggests that, once the pipeline is fully understood, its contribution is seen as non-trivial rather than purely “basic.”
>   - On **benchmarks and datasets**, Reviewer 1GWS notes that we “mainly contain two AVQA datasets only in the music domain.” In our rebuttal, we clarified that we actually evaluate on the **MUSIC-AVQA family and FortisAVQA**, which together form a canonical and increasingly standard ecosystem for audio-visual reasoning: they incorporate balanced splits, debiasing, robustness (rephrasings), and head–tail partitions. The focus of these benchmarks is not merely “music as a domain,” but the fact that they are **representative, high-quality testbeds for challenging audio-visual reasoning and long-tail AVQA**. Other reviewers also mentioned dataset scope, but did **not** treat it as a fatal flaw; they generally agreed this suite is a strong and widely used basis, and suggested multi-task extensions more as a future direction than as a rejection reason.
>   - On **baselines**, Reviewer 1GWS pointed out that methods like Qwen2.5-Omni, Kimi-Audio, and VideoSALMONN were initially missing. We have now **added exactly these models**, as well as QA-TIGER, CAT, and Meerkat, and the revised results show that EQUALS consistently achieves state-of-the-art or superior performance across MUSIC-AVQA and FortisAVQA. Other reviewers who raised baseline/comparison concerns (e.g., about Meerkat) indicated that, with these additions, their main reservations are resolved and their overall view of the paper is positive (e.g., 8g5K's score raised from 4 to 6).
>
>   Taken together, we believe Reviewer 1GWS’s concerns have been substantively addressed: the **novelty lies in the integrated question-guided pipeline and OT staging, which other reviewers acknowledge**, the **datasets are representative and standard for AVQA**, and **the requested baselines have been added with favorable outcomes**.
>
> We fully respect that the final decision must weigh all reviews. Our goal here is simply to assure you that we have taken the feedback seriously, substantially improved the clarity and experimental coverage, and that **EQUALS offers a principled and practically useful step forward** for question-guided audio–visual alignment and fusion.
>
> Thank you again for your consideration.
>
> Sincerely,
>  The Authors

---

### Meta-Review · Area_Chair_tTnW · 2026-01-07

**Summary:**

The reviewers provided mixed ratings. One reviewer evaluated the work positively, highlighting the question-guided alignment pipeline, the use of optimal transport, and adaptive fusion as meaningful contributions. However, the other reviewers raised several concerns: 1) Technical novelty: Multiple reviewers questioned whether the method introduces fundamentally new modeling ideas, or whether it mainly represents a careful system-level integration of existing components such as pooling, alignment, and MoE-based fusion. 2) Evaluation scope and fairness: Reviewers expressed concerns about limited dataset diversity, with experiments largely focused on MUSIC-AVQA variants, as well as the fairness and clarity of comparisons to strong prior methods. 3) Baselines and empirical rigor: Several reviewers requested more complete and up-to-date baselines, clearer baseline configurations, and stronger evidence that the reported gains are not primarily driven by the underlying LLM. 4) Presentation clarity.

Together, these issues led to a wide spread of initial scores, including one clear accept, two marginally below, and one reject.

**Reviewer Concerns:**

***Concerns Substantially Addressed by the Rebuttal***

Additional baselines:
The authors added several important and relevant baselines, including QA-TIGER, CAT, Meerkat, Qwen2.5-Omni, Video-SALMONN, and Kimi-Audio, improving the completeness of the experimental comparison.

OT loss ablation studies:
The rebuttal introduced controlled experiments that isolate the effects of question-guided pooling and optimal transport (OT) alignment. Additional explanations and ablations clarified the distinct functional roles of OT-before-pooling and OT-after-pooling.

Presentation and clarity:
Several organizational and presentation issues raised by reviewers were acknowledged and partially improved in the revised version, improving overall readability.

***Outstanding / Partially Addressed Concerns***

Technical novelty:
Although the rebuttal clarifies how existing components are integrated and staged, some reviewers may still view the contribution as primarily a system-level integration. This philosophical concern is likely to remain.

Fairness of the Meerkat comparison:
While Meerkat results were added, the reported performance on MUSIC-AVQA is substantially lower than what is reported in the original Meerkat paper. Since the dataset uses fixed splits, this discrepancy cannot be explained by differences in data partitioning, raising concerns about the fairness and consistency of the comparison.

Dataset scope:
Although the authors explained their focus on MUSIC-AVQA and its variants, the rebuttal does not fully address why evaluation on more general-domain AVQA datasets was not included. For example, Meerkat evaluates on an additional AVQA dataset beyond MUSIC-AVQA, which could have strengthened the empirical validation.

Task scope (why only AVQA):
Reviewer H3vC mentioned a concern regarding the exclusive focus on AVQA, particularly music-centric AVQA. If dataset availability is a limiting factor for AVQA, it remains unclear why the MLLM-based framework was not evaluated on related audio-visual tasks such as captioning, where larger and more diverse datasets across multiple domains are readily available. The authors simply frame this as future work in the rebuttal cannot address the concern.

**Reviewer Scores:**

Reviewer gRbU (initial score: 8):
This reviewer was strongly positive about the technical design and contributions, highlighting the question-guided alignment pipeline, optimal transport, and adaptive fusion. Their questions were largely about clarification, which were addressed in the rebuttal. The score would very likely remain at 8.

Reviewer 8g5K (initial score: 4 → updated to 6):
This reviewer explicitly stated in the discussion that the rebuttal addressed the majority of their concerns and raised their score to 6. No further change is expected beyond this update.

Reviewer H3vC (initial score: 4):
This reviewer viewed the work as well executed, with the main request being additional comparisons (notably Meerkat). While the authors added these baselines, concerns remain regarding the fairness and consistency of the Meerkat comparison relative to reported results in the original paper and the limited dataset scope. As a result, the score would likely remain at 4.

Reviewer 1GWS (initial score: 2):
This reviewer expressed strong skepticism regarding novelty, dataset scope, and missing baselines. Although some concrete issues (e.g., missing baselines, presentation) were addressed, the reviewer’s core objection about limited novelty and narrow evaluation domain appears philosophical and unlikely to change substantially. The score would likely remain at 2, or at best increase slightly but still below the acceptance threshold.

The AC sees no clear indication that reviewer scores would converge toward acceptance even after full discussion. While one reviewer remains strongly positive and another upgraded to an accept-level score, two reviewers would likely remain below threshold due to the outstanding concerns about novelty, evaluation scope, and fairness of comparisons. The AC concurs with these remaining concerns.

---

### Decision · Program_Chairs · 2026-01-26

Reject